# Deep learning boosts sensitivity of mass spectrometry-based immunopeptidomics

Mathias Wilhelm [1,2,13✉], Daniel P. Zolg[2,13], Michael Graber[2], Siegfried Gessulat [2], Tobias Schmidt [2], Karsten Schnatbaum[3], Celina Schwencke-Westphal [4,5,6], Philipp Seifert[4,6], Niklas de Andrade Krätzig[6,7,8], Johannes Zerweck[3], Tobias Knaute[3], Eva Bräunlein[4,6], Patroklos Samaras [2], Ludwig Lautenbacher [2], Susan Klaeger [9], Holger Wenschuh[3], Roland Rad [6,7,8], Bernard Delanghe[10], Andreas Huhmer[11], Steven A. Carr [9], Karl R. Clauser[9], Angela M. Krackhardt [4,5,7], Ulf Reimer[3] & Bernhard Kuster [2,12✉]

Characterizing the human leukocyte antigen (HLA) bound ligandome by mass spectrometry (MS) holds great promise for developing vaccines and drugs for immune-oncology. Still, the identification of non-tryptic peptides presents substantial computational challenges. To address these, we synthesized and analyzed >300,000 peptides by multi-modal LC-MS/MS within the ProteomeTools project representing HLA class I & II ligands and products of the proteases AspN and LysN. The resulting data enabled training of a single model using the deep learning framework Prosit, allowing the accurate prediction of fragment ion spectra for tryptic and non-tryptic peptides. Applying Prosit demonstrates that the identification of HLA peptides can be improved up to 7-fold, that 87% of the proposed proteasomally spliced HLA peptides may be incorrect and that dozens of additional immunogenic neo-epitopes can be identified from patient tumors in published data. Together, the provided peptides, spectra and computational tools substantially expand the analytical depth of immunopeptidomics workflows.

[1] Computational Mass Spectrometry, Technical University of Munich (TUM), Freising, Germany. [2] Chair of Proteomics and Bioanalytics, Technical University of Munich (TUM), Freising, Germany. [3] JPT Peptide Technologies GmbH, Berlin, Germany. [4] Klinik und Poliklinik für Innere Medizin III, Klinikum rechts der Isar, School of Medicine, Technical University of Munich (TUM), Munich, Germany. [5] German Cancer Consortium (DKTK), partner site Munich; and German Cancer Research Center (DKFZ), Heidelberg, Germany. [6] Center for Translational Cancer Research (TranslaTUM), TUM School of Medicine, Technical University of Munich (TUM), Munich, Germany. [7] Institute of Molecular Oncology and Functional Genomics, TUM School of Medicine, Technical University of Munich (TUM), Munich, Germany. [8] Klinik und Poliklinik für Innere Medizin II, Klinikum rechts der Isar, School of Medicine, Technical University of Munich (TUM), Munich, Germany. [9] Broad Institute of MIT and Harvard, Cambridge, MA, USA. [10] Thermo Fisher Scientific, Bremen, Germany. [11] Thermo Fisher Scientific, San Jose, CA, USA. [12] Bavarian Biomolecular Mass Spectrometry Center (BayBioMS), Technical University of Munich (TUM), Freising, Germany. [13] These authors contributed equally: Mathias Wilhelm, Daniel P. Zolg. ✉email: mathias.wilhelm@tum.de; kuster@tum.de

dentifying neo-epitopes in the leukocyte antigen (HLA) ligandome of cancer patients has become a major source for potential therapeutic intervention[1,2]. Direct evidence for the presentation of disease- or patient-specific HLA antigens on tumor cells can only be obtained by measuring HLA-bound ligands. Mass spectrometry (LC-MS/MS) is the primary method for this purpose[3,4]. However, compared to the analysis of (mostly) tryptic peptides in proteomics, the analysis of HLA peptides poses substantial challenges. These arise from the fact that HLA peptides are generated by unspecific protease cleavage. This not only alters the characteristics of the tandem mass spectra (i.e., the type and intensity of the fragment ions observed), but also vastly increases the number of sequences a search engine has to consider as a possible match to a particular MS/MS spectrum[5]. As a result, commonly used search engines do not comprehensively and confidently identify non-tryptic peptides and many naturally occurring HLA peptides may, therefore, be missed, diminishing the chances of discovering therapeutically relevant antigens[6]. At least in part, this is because search engine scoring schemes are biased toward tryptic peptides and do not make use of the intensities of fragment ions. We and others have recently shown that including intensity-based matching scores can substantially increase the number of confidently identified tryptic peptides[7–9] but none of the currently available predictors have been trained on (non-tryptic) HLA peptides. Here we show, that expanding our deep learning framework Prosit[7] to the interpretation of non-tryptic peptides also greatly improves the identification of HLA peptides and neo-epitopes and we demonstrate that proteasomal splicing of peptides is much rarer than anticipated.

## Results

**Extension of the ProteomeTools libraries of synthetic peptides and mass spectra to non-tryptic peptides.** The number of theoretically possible HLA bound (neo)-antigens is orders of magnitude larger than generated by common proteomic workflows using e.g., trypsin[6]. Therefore, no collection of empirically observed HLA peptides and their LC-MS/MS spectra can (likely) ever be complete. Essentially, this precludes the use of such resources for comprehensive HLA peptide identification in any particular sample. Instead, we reasoned that generating high-quality LC-MS/MS data for a representative set of synthetic peptides with precisely known sequences may form the basis for the computational prediction of spectra and chromatographic retention times for any peptide. Within the ProteomeTools project[10], we synthesized ~305,000 non-tryptic peptides comprising ~169,000 HLA class I, ~73,000 HLA class II, ~32,000 AspN, and ~31,000 LysN sequences and measured all of these using 11 different LC-MS/MS parameters (Fig. 1a, median synthesis/detection success 88%; Supplementary Fig. S1a–c and Supplementary Data 1). The HLA sequences were selected from published HLA ligandomes[11–13] (Fig. 1b) and AspN/LysN peptides were drawn from a large unpublished study (kindly provided by Josh Coon, Univ. of Wisconsin). Together with previously published tryptic peptides and MS data, the ProteomeTools resource now contains ~100 million high-quality tandem mass spectra (Andromeda score >100) from 1.6 million unique peptide precursors (combination of sequence, charge state, and modifications), covering nearly all possible N-and C-terminal amino acid combinations with the exception of peptides comprising of cysteine residues at the N- and C-terminus which are underrepresented in all sampled resources (Fig. 1c).

Non-tryptic peptides often exhibit LC-MS/MS characteristics very distinct from tryptic peptides, both in terms of chromatography (Supplementary Fig. S1d) and peptide fragmentation. For instance, strong internal ion series and neutral losses are often

observed and the localization of basic amino acid residues within the peptide sequence often directs fragmentation in an (at first sight) unpredictable way. An extreme case is shown in Fig. 1d, where the assignment of the experimental spectrum (upper panel) to the peptide sequence TSGYGQSSYSSY (gene *FUS*) may raise concerns even for a trained eye. However, the spectra collected from a tumor-derived HLA preparation[11] or from the synthetic peptide agree extremely well (spectral contrast angle, SA = 0.9), validating their identity.

Due to the differences in fragmentation characteristics of tryptic and non-tryptic peptides, there is an ongoing debate how to best acquire LC-MS/MS data for HLA peptides[14]. The data presented here is well suited for addressing this question because every synthetic peptide was fragmented by up to 11 different LC-MS/MS parameters (including 6 different collision energy settings for HCD), resulting in ~24 million high-quality reference spectra (Andromeda score >100; Fig. 1e, Supplementary Fig. S2a). Overall, CID and HCD fragmentation with Orbitrap or ion trap detection achieved the highest scores and score distributions resemble those of tryptic peptides (Supplementary Fig. S2b). Not surprisingly, ETD and combined scan modes generally did not work well particularly for the short and mostly doubly charged HLA Class I peptides. In contrast, HLA class II, AspN and LysN peptides with higher charges also resulted in high scoring ETD spectra. Compared to data of endogenous peptides from a large publication[11], especially synthetic HLA class II peptides showed higher Andromeda scores (Supplementary Fig. S2c). The use of ion trap detection (IT) showed no linear correlation between identification scores and fragment ion intensity ($R = 0.02$). In contrast, Orbitrap detection required stronger signals to achieve high scores ($R = 0.33$, Supplementary Fig. S2d). One might, therefore, argue that IT detection is better suited for the analysis of low abundance peptides as often encountered in clinical samples. However, direct comparisons of different fragmentation modes (Supplementary Fig. S3 for HLA Class I peptides, Supplementary Fig. S4 for a global analysis and other sets) corroborated previous notions that combining several types of fragmentation and mass analyzers in a single analysis can offer advantages[15] as no single fragmentation mode and mass analyzer combination alone was superior in the majority of cases.

While public resources such as SysteMHC[13] also contain datasets employing a range of fragmentation settings (Supplementary Fig. S5a), there are substantial differences across the datasets collected nominally using the same normalized collision energies (NCE; instrument to instrument variation) and chromatographic retention times (differences between stationary phases and gradients; Supplementary Fig. S5b-d, Supplementary Data 2 and Supplementary Notes). These inconsistencies impair the use of published data for e.g., spectral library searching or machine learning which is why all following work is solely based on data acquired in the ProteomeTools project.

**A single Prosit model allows accurate prediction of tryptic and non-tryptic peptide MS/MS spectra.** Combining the 9 million HCD spectra of non-tryptic peptides collected in this study with the 21 million previously published tryptic peptide spectra[7,10] enabled the training (70% of all spectra), testing (20%), and validation (10%, referred to as holdout set) of a single high-performant Prosit model for both types of peptides (Fig. 2a, Supplementary Fig. S6a, "Methods"). Comparison of the previously published (HCD Prosit 2019) and the here developed (HCD Prosit 2020) Prosit models showed a substantially improved normalized spectral contrast angle (SA, Supplementary Fig. S6b) between predicted and experimental spectra for non-tryptic peptides (SA > = 0.9 for 18% vs 43% of spectra, respectively) and a

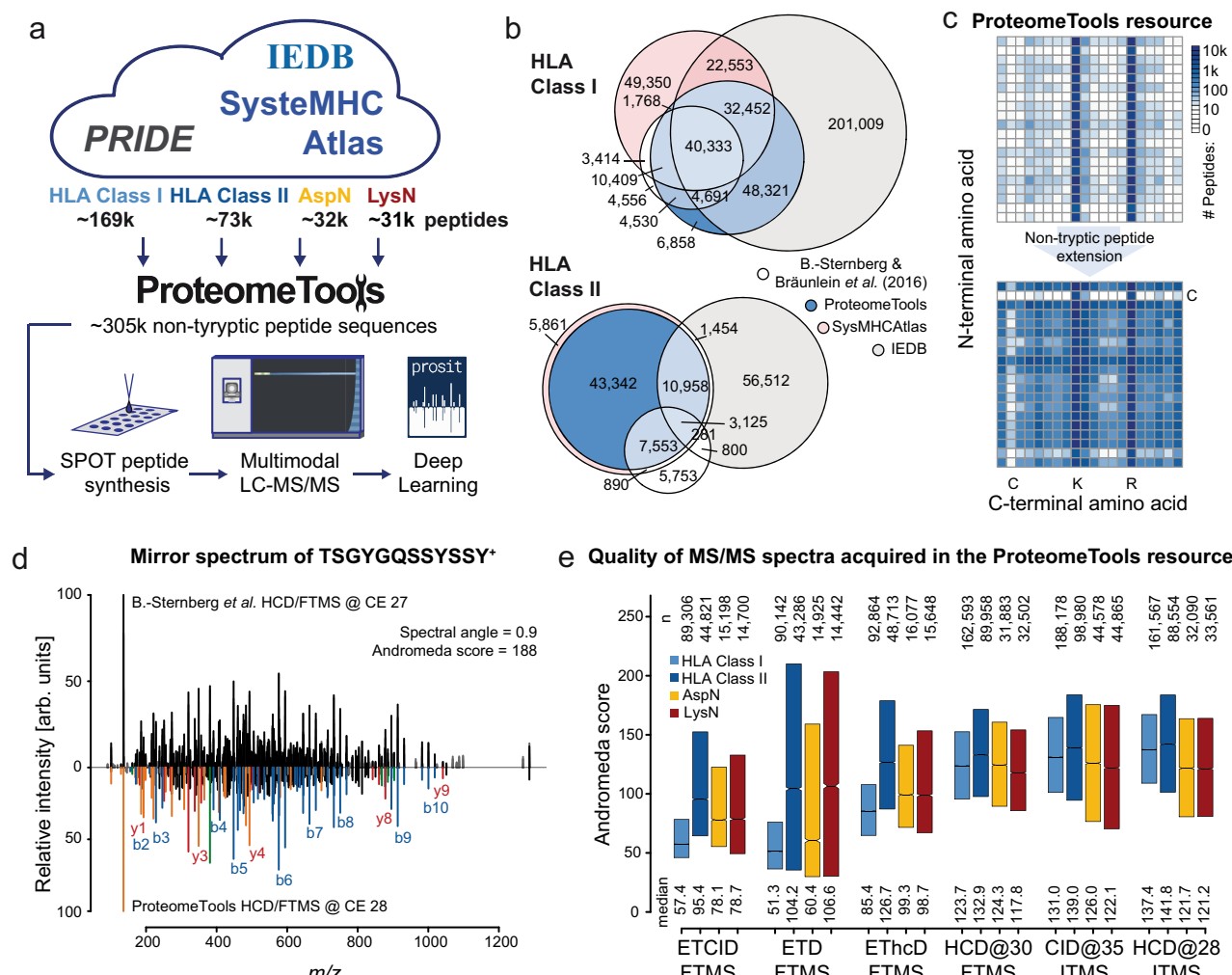

**Fig. 1 ProteomeTools' non-tryptic peptide extension. a** The ProteomeTools resource was extended in this study by ~305 k non-tryptic synthetic peptides consisting of ~169 k HLA class I, ~73 k HLA class II, ~32 k AspN and 31 k LysN peptides. All peptides were systematically characterized by multimodal LC-MS/MS. All data were subsequently used for training the 2020 Prosit fragment intensity prediction model. **b** Proportional Venn diagrams of HLA class I (top) and HLA class II (bottom) peptides in ProteomeTools (blue), SysteMHC Atlas (light red), IEDB (gray), and B.-Sternberg et al.[11]. (white). **c** Number of peptides (log10 color scale, white to dark blue) synthesized for the ProteomeTools resource sorted by N- (y-axis) and C-terminal (x-axis) amino acid without (top) and with (bottom) the extension of non-tryptic peptides from this study. **d** Mirror spectrum of the singly charged non-tryptic peptide TSGYGQSSYSSY acquired by B.-Sternberg and Bräunlein et al.[11] (endogenous peptide, top) and ProteomeTools (synthetic peptide, bottom). Fragment ion peaks with and without neutral losses are annotated in blue, red, green, and orange for b-, y-, a- and internal fragment ions, respectively. The spectral similarity measured by the normalized spectral contrast angle between the two spectra and the Andromeda matching score of the top spectrum are shown in the top. **e** Boxplots of Andromeda scores for the best MS/MS identification per precursor for HLA class I (dark blue), HLA class II (light blue), AspN (yellow), and LysN (red) peptides for different fragmentation settings (HCD, CID, ETD, EThcD, ETciD) and mass analyzers (FTMS: Orbitrap mass analyzer, ITMS: ion trap mass analyzer). The number of spectra (n) and median Andromeda score (median) are depicted at the top and bottom of the boxplot. The box indicates the interquartile range (IQR). The black line marks the median, notches extend to 1.58 * IQR/sqrt(n), no whiskers or outliers outside IQR shown. Raw and analysis data are available from the PRIDE repository with identifier PXD021013.

slight improvement for tryptic peptides (SA >= 0.9 for 46% vs 50%; Fig. 2b). The 2020 model performed drastically better for singly charged peptides (median SA of 0.89 vs 0.56; Fig. 2c) and also improved prediction accuracy for higher charged peptides (Supplementary Fig. S7a), across different C-terminal amino acids (Supplementary Fig. S7b, c) as well as peptides with internal or N-terminal basic residues (Supplementary Fig. S7d, e). The 2020 model also performed well across the different peptide sets, precursor charges, and the wide range of collision energies used for training (Supplementary Fig. S8a–c) and showed little bias for C-terminal amino acids (Supplementary Fig. S8d) and combinations of peptide N- and C-termini (Supplementary Fig. S9). This demonstrates that the 2020 Prosit model is well suited for the

analysis of both tryptic and non-tryptic peptides. A representative example is shown in Fig. 2d (YPYPVSNSV; $[M + H]^+$; gene ATXN2L). It is apparent that the experimental and predicted fragment ion intensities of the annotated b- and y-ions are in very good agreement (SA = 0.88). Perhaps surprisingly, no major biases were observed when comparing measured and predicted b- and y- fragment ion intensities even for peptides that show dominant internal or neutral loss ion series (Fig. 1d; Supplementary Fig. S10 and Supplementary Notes).

Because CID fragmentation can be valuable for the analysis of non-tryptic peptides (see above), and the fact that linear ion trap instruments are widely used in the proteomic community, we also trained a dedicated Prosit CID model using the CID ITMS

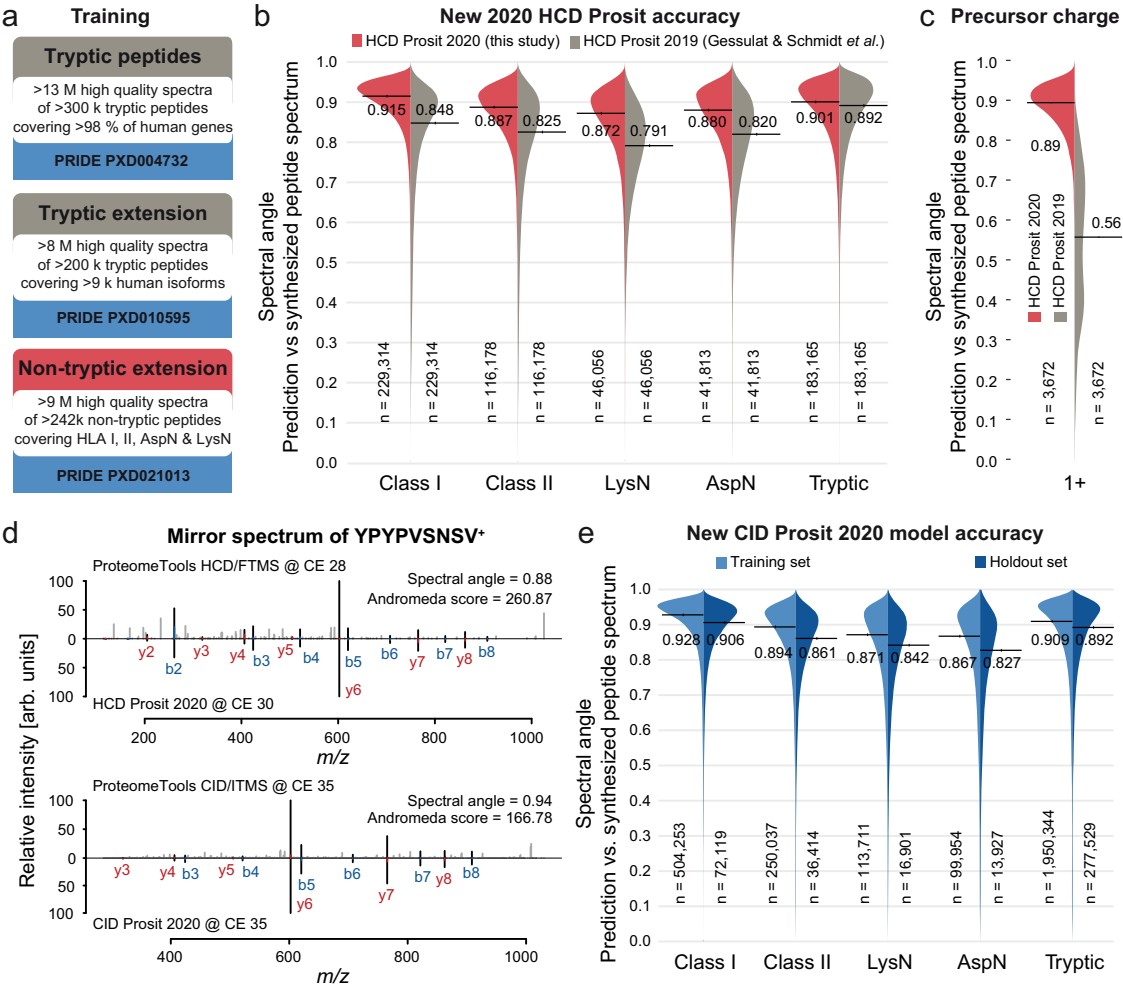

**Fig. 2 Deep learning framework Prosit for tryptic and non-tryptic peptide fragment intensity prediction. a** The deep learning framework Prosit was trained on data available prior to this study (tryptic peptides, top panel; tryptic extension, middle panel) and data on non-tryptic peptides (bottom panel) generated in this study. **b** Beanplots comparing the prediction accuracy of the HCD Prosit 2020 model (red, this study) against the prediction accuracy of the previously published HCD Prosit 2019 model (gray, tryptic only, Gessulat and Schmidt et al.[7]) for the four introduced peptides sets (HLA class I, HLA class II, LysN, and AspN) and the previously published tryptic peptides. The number of underlying spectra (n) is indicated at the bottom. The black solid line and corresponding numbers indicate the median spectral angle for each distribution. **c** Beanplot comparing the prediction accuracy of the HCD Prosit 2020 model (red, this study) against the HCD Prosit 2019 model (gray, Gessulat and Schmidt et al.[7]) for singly charged peptides. The number of underlying spectra is indicated at the bottom. The black solid line and corresponding numbers indicate the median spectral angle for each model. **d** Mirror spectrum of the singly charged non-tryptic peptide YPYPVSNSV comparing the experimentally acquired HCD ProteomeTools spectrum (top panel, top spectrum) to its predicted spectrum by the HCD Prosit 2020 model (top panel, bottom spectrum) and the experimentally acquired CID ProteomeTools spectrum (bottom panel, top spectrum) to its predicted spectrum by the CID Prosit 2020 model (bottom panel, bottom spectrum). Fragment ions are labeled in blue and red for b- and y-ions, respectively. Matching peaks (present in both spectra) are visualized in black, whereas peaks only present in the top (experimental) spectra are colored in gray. Red and blue fractions of matching peaks indicate the normalized difference in intensity between the experimental and predicted spectra. **e** Beanplots comparing the prediction accuracy of the CID Prosit 2020 model between the training (light blue) and holdout (dark blue) set for the four introduced peptides sets (HLA class I, HLA class II, LysN, and AspN) and previously published tryptic peptides. The number of underlying spectra (n) is indicated at the bottom. The black solid line and corresponding numbers indicate the median spectral angle for each distribution. Raw and analysis data are available from the PRIDE repository with identifiers PXD004732, PXD010595, and PXD021013.

dataset. Exemplified by the mirror plot shown in Fig. 2d (lower panel; SA = 0.92) the CID model showed similar performance as the HCD model for all types of peptides (Fig. 2e; Supplementary Fig. S11). No adaptations to the Prosit indexed retention time (iRT) 2019 model for the prediction of chromatographic retention times were necessary. It, therefore, appears that Prosit had already learned a generic representation of peptide hydrophobicity such that the old model could be directly applied to non-tryptic peptides and leading to outstanding agreement between measured and predicted iRT values (ΔiRT 95% = 4.68, Pearson $R > 0.99$, Supplementary Fig. S6b). To demonstrate the

practical utility and impact of the Prosit models developed here, the following sections provide three pertinent use cases.

**Prosit boosts the number of identified HLA peptides from human cell lines.** Akin to what was previously observed for tryptic peptides, we hypothesized that the integration of fragment ion intensity predictions into the database matching process should lead to an improvement in the number of HLA peptide identifications. To test this, we re-processed data from a recent study that analyzed HLA class I peptides from 95 monoallelic cell

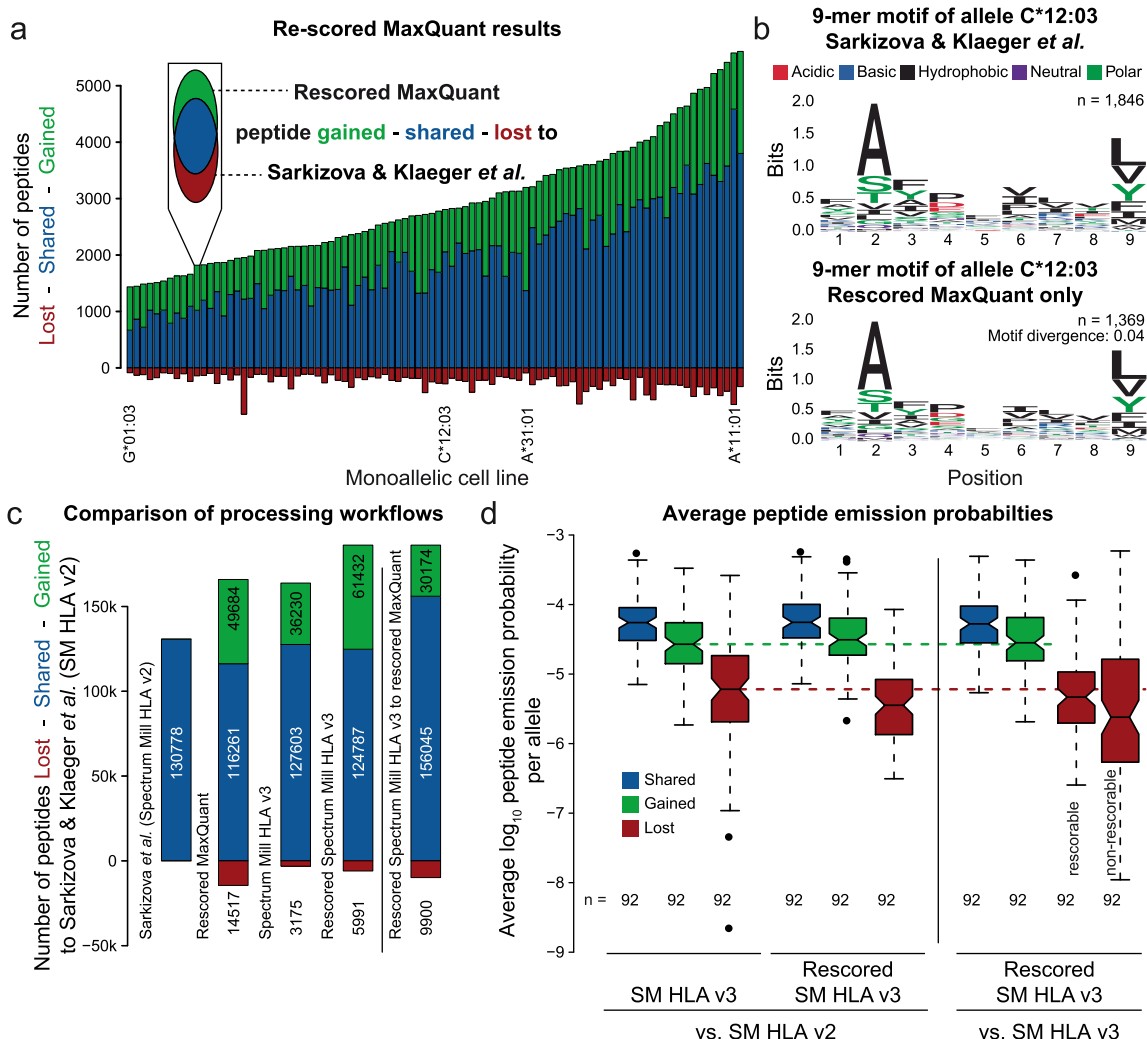

**Fig. 3 Deep learning-assisted rescoring of data from 92 monoallelic cell lines. a** Vennbars (simplification of Venn diagram as depicted in the inset) showing the number of peptides lost (red), shared (blue) and gained (green) from Prosit-based rescoring of MaxQuant results of data published by Sarkizova and Klaeger et al.[16] (using Spectrum Mill HLA v2) for each of the 92 monoallelic cell lines investigated in this study. **b** Peptide motif plot of 1846 9-mers confidently identified to be present in the cell line expressing allele C*12:03 from the published results (top panel) and 1369 9-mers added by the Prosit-based rescoring of MaxQuant results (bottom). Amino acids are colored according to their physio-chemical properties (black denotes hydrophpbic, red acidic, blue basic, purple neutral, and green polar amino acids). The difference between the motifs was estimated by Jensen-Shannon divergence (indicated in the bottom motif) comparing the positional weight matrices of both motifs. **c** Vennbars showing the number of peptides lost (red bar), shared (blue), and gained (green) when comparing results obtained from different workflows to results published by Sarkizova and Klaeger et al.[16] (left of solid vertical line). The bar to the right of the vertical line compares the number of lost, shared, and gained peptides when comparing the Prosit-based rescored results of Spectrum Mill HLA v3 to the Prosit-based rescored results of MaxQuant. **d** Boxplots of the average emission probabilities (probability that a peptide is derived from a certain motif, see "Methods") per allele of peptides shared (blue), gained (green) and lost (red) when comparing results obtained from the workflows indicated at the bottom of the plot. The number of alleles (n) for which an average emission probability was calculated is depicted at the bottom. Peptides lost (not confidently identified) by the rescored SM HLA v3 workflow in comparison to the SM HLA v3 workflow are shown separately depending on whether the fragment intensities of the peptide could be predicted by Prosit (rescorable) or not (non-rescorable). The box indicates the interquartile range (IQR). The black line marks the median, notches extend to 1.58 * IQR/sqrt(n), whiskers to 1.5 * IQR from the hinge. Data outside whiskers are plotted individually as black dots. Raw and analysis data are available from the PRIDE repository with identifier PXD021398 and MassIVE repository with identifiers MSV000084172 and MSV000080527.

lines[16] using MaxQuant and re-scored all proposed peptide spectrum matches (PSMs) by integrating Prosit's fragment intensity predictions ("Methods"; Supplementary Data 3). Comparing the results to those obtained by Spectrum Mill (SM) HLA v2 (Fig. 3a), a search engine that is optimized for the identification of HLA peptides and that was used in the aforementioned study, showed an average improvement by a factor of 1.5 (green bars) across all alleles and only marginal losses (red bars). The alleles A*31:01 (C-terminal arginine) and G*01:03 (internal

prolines and C-terminal leucine) exhibited the largest increase by a factor of 2 whereas the allele A*11:01 (C-terminal lysine and arginine) showed the least increase, further corroborating that Prosit's prediction accuracy is well suited for the analysis of both tryptic and non-tryptic peptides. The observed improvement could be solely attributed to rescoring MaxQuant results by integrating intensity information of the fragment ions, as standalone MaxQuant showed substantially fewer peptide identifications (Supplementary Fig. S12a–e).

This monoallelic dataset allowed us to investigate the allele-specific prediction accuracy and extent of possible overfitting of the 2020 Prosit models by mapping our holdout dataset to the results obtained by Sarkizova and Klaeger et al.[16]. The results show that, on average, a spectral angle of 0.92 was achieved without substantial bias for any allele (Supplementary Fig. S13a), further supporting that the 2020 Prosit model can accurately predict fragment intensities. To further validate the earlier observation that the 2020 Prosit model was substantially improved particularly for singly charged peptides, we rescored the allele C*08:02 showing a high prevalence of singly charged peptides using the 2019 and 2020 model. While the 2019 model already led to a net improvement of ~300 peptides when rescoring MaxQuant results, the 2020 model shows substantially fewer losses (70 vs ~200 using the 2019 model) and led to a net increase of ~850 peptides (Supplementary Fig. 13b).

In order to validate that peptides added by rescoring MaxQuant results are likely true binders, we compared their binding motifs to those from the original study. As an example, the extracted 9mer peptide motifs of allele C*12:03 from the original study (Fig. 3b top panel) and the peptides added by the rescored MaxQuant results (Fig. 3b bottom panel) are almost indistinguishable (Jensen-Shannon divergence of 0.04; "Methods"). No major differences were observed when investigating the sequence motifs of all 8–11mers for any of the 92 alleles suggesting that the rescoring workflow retains peptides that are likely true HLA binders (Supplementary Data 4).

The original study reported the identification of ~131,000 unique peptides for the 92 alleles reprocessed here (Fig. 3c). Prosit rescoring of MaxQuant retained ~116,000 of these peptides and added nearly 50,000 further peptides (38%) resulting in a net increase of ~35,000 peptides (27%). The loss of ~15,000 peptides (11%) is attributed to the difference in data processing by MaxQuant and SM HLA v2 since the majority of these were not on the candidate list of MaxQuant and thus could not be rescored by Prosit (Supplementary Fig. S13c).

In follow up to some of the above observations, the dataset was re-processed by Spectrum Mill using further optimized HLA scoring (referred to as SM HLA v3; Supplementary Notes) which led to a net increase of ~33,000 peptides (25%) without major losses when compared to SM HLA v2 (Fig. 3c). Next, the SM HLA v3 results were rescored using Prosit which added further 23,000 peptides (17%) to the published results (combined net total of 55,000 additional peptides). Comparing rescored MaxQuant results to rescored SM HLA v3 results (Fig. 3c, last bars) showed that, although SM HLA v3 rescoring outperformed MaxQuant rescoring, the gains and losses are in the range of what one might expect from using different search engines.

Peptides added by either SM HLA v3 or rescored SM HLA v3 again exhibited the expected peptide binding motif across all alleles and peptide lengths (Supplementary Data 4). More specifically, the motifs of gained peptides were much more similar to those of the original set than the lost peptides (assessed by the motif-dependent emission probability; Fig. 3d, left panel; "Methods"; Supplementary Fig. S13f). This further corroborates that Prosit rescoring retains more potential true binders while rejecting more potential non-binders (false positives). Because the current Prosit HLA model was not trained on peptides containing free cysteine side chains or other amino acid modifications that may be identified on HLA peptides, 3912 peptides were lost because they could not be rescored (Fig. 3d, right panel). These lost peptides show a similar emission probability distribution compared to lost rescorable peptides (12,212 peptides). This does not mean that these specific cases are necessarily false positives, because neither motif generation nor emission probability calculation take the modification status of peptides into account.

Therefore, other factors might be responsible for their apparently reduced emission probabilities. Taken together, re-processing the 92 monoallelic cell lines using SM HLA v3 and subsequent Prosit rescoring identified, on average, 58% more peptides per allele and led to the identification of a total of ~186,000 peptides (42%), attesting to the substantial gains that may be made in the analysis of HLA peptides from cell lines when fragment ion intensities are incorporated into the identification process.

**Prosit rescoring questions the prevalence of proteasomal splicing of peptides.** The high prediction accuracy of Prosit provided an opportunity to investigate the much debated prevalence of HLA peptides that originate from splicing events during proteolysis within the proteoasome[17–19]. We retrieved the identification results (PSMs) from the seven HCD raw files[20] from which 1230 spliced peptides (23.6% of all identified HLA ligands) were identified in a prior analysis[21] using Mascot and predicted their spectra (8329 non-spliced and 1588 spliced PSMs) using Prosit. When comparing the spectral angle distributions of the reported non-spliced (canonical) peptides to spliced peptides (Fig. 4a), it is apparent that most of the proposed spliced peptides have much lower spectral similarity to the Prosit predictions compared to the canonical peptides (SA = 0.72 vs SA = 0.87) and 23.3% vs 2.4% of the PSMs have SA values lower than 0.5, respectively. The latter in particular suggests that a large proportion of the proposed spliced peptides may not be correct.

To investigate this further, the raw MS data were also re-analyzed using MaxQuant and MSFragger and rescored by Prosit (Fig. 4b) using a database consisting of canonical peptides only to ask if the spectra assigned to spliced peptides may be better explained by canonical sequences. To be able to compare the results obtained from the three different search engines (original Mascot, re-analyzed MaxQuant, and MSFragger), we rescored all PSMs of each search engine separately, but trained a single Percolator model[22] using only the features calculated by the rescoring pipeline (Fig. 4b, red arrows). This allowed to reassess and estimate the global FDR of the Mascot results of the original study, as the original results were filtered by the Mascot ion score only. More importantly, it also allowed the ranking of the best PSM from each of the three search engines for every spectrum ("Methods"; Supplementary Notes).

Comparing the results from the three different search engines (Supplementary Fig. S14a) revealed that while spectra annotated as spliced peptides showed very high agreement between predicted and experimental fragment intensities, the same spectra can also be confidently matched to canonical peptides found by rescored MaxQuant and/or MSFragger results (Supplementary Data 5–6). Figure 4c shows an example where the spliced peptide proposed in the original study using Mascot (top panel, splice position indicated by the pipe symbol) and the canonical peptide proposed by rescored MaxQuant and rescored MSFragger (bottom panel) only differ by the isobaric amino acid combination of GVA and NL in the 3 C-terminal amino acids (Levenshtein distance of 3). The spectral angles between the experimental and both predicted spectra are the same (SA = 0.95) and the Percolator score difference between the proposed spliced (Score = 2.9) and canonical peptide (Score = 2.5) is small. In other words, there is essentially no information in the experimental or predicted spectra to indicate that either sequence may be the more confident match. Thus, we propose that the canonical peptide should be the favorable hypothesis over a proteasomal splicing event.

In total, the re-analysis indicates that 1067 of the 1,230 (87%) proposed spliced peptides are not conclusively supported by the mass spectrometry data (Fig. 4d). This is because either (i) they

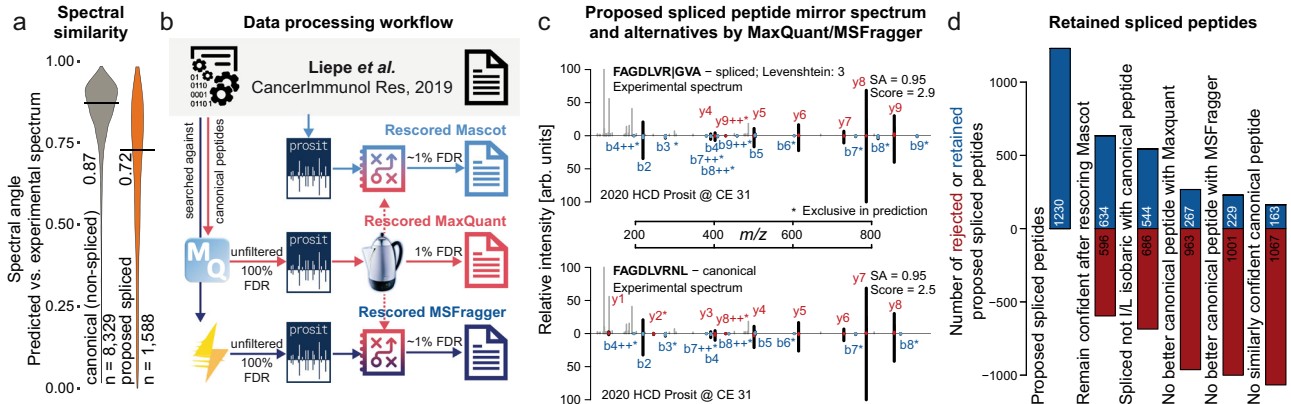

**Fig. 4 Re-assessment of HLA peptides proposed to be generated by splicing. a** Spectral angle distributions comparing HCD Prosit 2020 predicted spectra against experimentally acquired spectra of non-spliced (gray) and proposed spliced (orange) peptides extracted from Liepe et al.[21]. **b** Identification results by Mascot from the original study were retrieved and rescored with Prosit. The raw MS data were also retrieved, re-searched by MaxQuant and MSFragger and rescored using Prosit. A single Percolator model was trained for confidence estimation, based on the results obtained from the MaxQuant analysis. This model was applied to the data from the rescored Mascot and rescored MSFragger results. **c** Two mirror plots of an experimental spectrum (top spectrum in both) identified either as the proposed spliced peptide FAGDLVR|GVA (top mirror plot, pipe symbol indicates splicing position) or non-spliced peptide alternative FAGDLVRNL (bottom mirror plot) plotted against the corresponding HCD Prosit 2020 predicted spectrum (bottom spectrum each). The spliced and non-spliced peptide sequences differ in 3 amino acids (Levenshtein distance 3). The spectral angle (SA) compares the predicted b- and y-ion intensities to the corresponding matching peaks in the experimental spectrum (excluding any observed but not matched peaks in the experimental spectrum). Matching fragments are highlighted in black whereas peaks without match are shown in gray. Fragment ions which are exclusively present in the predicted spectrum are marked with an asterisk (*). The blue and red fractions of the matched b- and y-ions (respectively) indicate the normalized intensity difference of these fragments. The confidence score (Score) of the proposed match was estimated by the shared Percolator model and is indicated at the top. **d** Barplots of different consecutive filtering steps of retained (blue) and rejected (red) proposed spliced peptides by various quality control steps. Raw and analysis data are available from the PRIDE repository with identifiers PXD021398, PXD000394, and Mendeley with identifier y2cvb5nvgn.1.

did not remain confident when using the Prosit-based rescoring pipeline (rejecting 596 spliced peptides, 48%), or (ii) the spliced and canonical peptides are I/L isomers (90 spliced peptides, 7%) that cannot be distinguished by mass spectrometry or (iii) a more confident canonical PSM was identified by MaxQuant and/or MSFragger (315 spliced peptides, 26%), or (iv) the proposed spliced peptide did not have a substantially better percolator score supporting its identification over a canonical peptide (66 peptides, 5%; also see the example in Fig. 4c, Supplementary Fig. S14, and Supplementary Notes). A similar analysis performed for the 3994 reported canonical peptides from the same study rejected just 179 (4%) peptides because they did not remain confident after rescoring the Mascot results and another 296 peptides (7%) because a more confident canonical PSMs was identified by MaxQuant and/or MSFragger.

Together, the re-analysis of the data indicates that only 3% (instead of the published 23%) of the HLA peptides in the original study may represent genuine spliced peptides. However, there may well be additional factors not investigated here which may reduce the number of candidate spliced peptides further. For example, as previously observed[23] the predicted and observed retention time (RT) of the proposed spliced peptides differ substantially from non-spliced peptides. We performed a similar analysis using Prosit and the results show that the retained proposed spliced peptides exhibit substantially larger than expected RT deviations (Supplementary Fig. 14d). Re-analysis of a database collecting proposed spliced peptide sequences for community use[19], showed similar signs of inflated numbers of proposed spliced peptides. Also here, spliced peptides showed lower identification scores compared to non-spliced peptides (Supplementary Fig. 14e).

Our findings agree with previous studies that criticized the identification of proteasomal splicing of HLA peptides[23,24] and, more broadly suggest that additional evidence is required for the

unambiduous identification of spliced HLA peptides. While Prosit shows very high overall prediction accuracy, it is possible that it makes mistakes for some sequences and which could include HLA peptides. Here, spectra of sysnthetic peptides of proposed spliced peptides may be used to find such prediction inaccuracies and lend more support to the identification of spliced HLA peptides. However, spectra of synthetic peptides will not always be conclusive because finding appropriate spectrum similarity thresholds between spectra of synthetic and endogenous peptides is not trivial. While our analysis does not rule out that proteasomal splicing exists and that this may represent important biology, it currently appears to be much less prevalent than anticipated.

**Prosit rescoring increases the chances of finding (immunogenic) neoepitopes in cancer patients.** As a third case, we investigated the potential of Prosit rescoring for the detection of neoepitopes in cancer patients. Strikingly, re-processing of published HLA Class I and Class II immune peptidome data from 25 human melanoma patients[11] identified on average 3-fold more HLA class I peptides per patient (Fig. 5a; Supplementary Fig. S15a) and on average 2.4-fold for HLA class II peptides (Supplementary Fig. S15b, c). This is a stronger gain than observed for the cell lines above likely because the limited sample quantities obtained from patients led to weaker spectra. These benefit from rescoring more strongly than those collected from cell lines (Supplementary Fig. S15d). To find out if such extended patient immunopeptidomes would translate into higher chances of finding patient-specific neoepitopes, we reanalyzed the HLA class I raw data from patient Mel15 using a patient-specific protein sequence database containing its genomic alterations (mutations, inserts, deletions, or frameshifts). Rescoring identified 78 neo-epitopes (Fig. 5b, "Methods") substantially surpassing

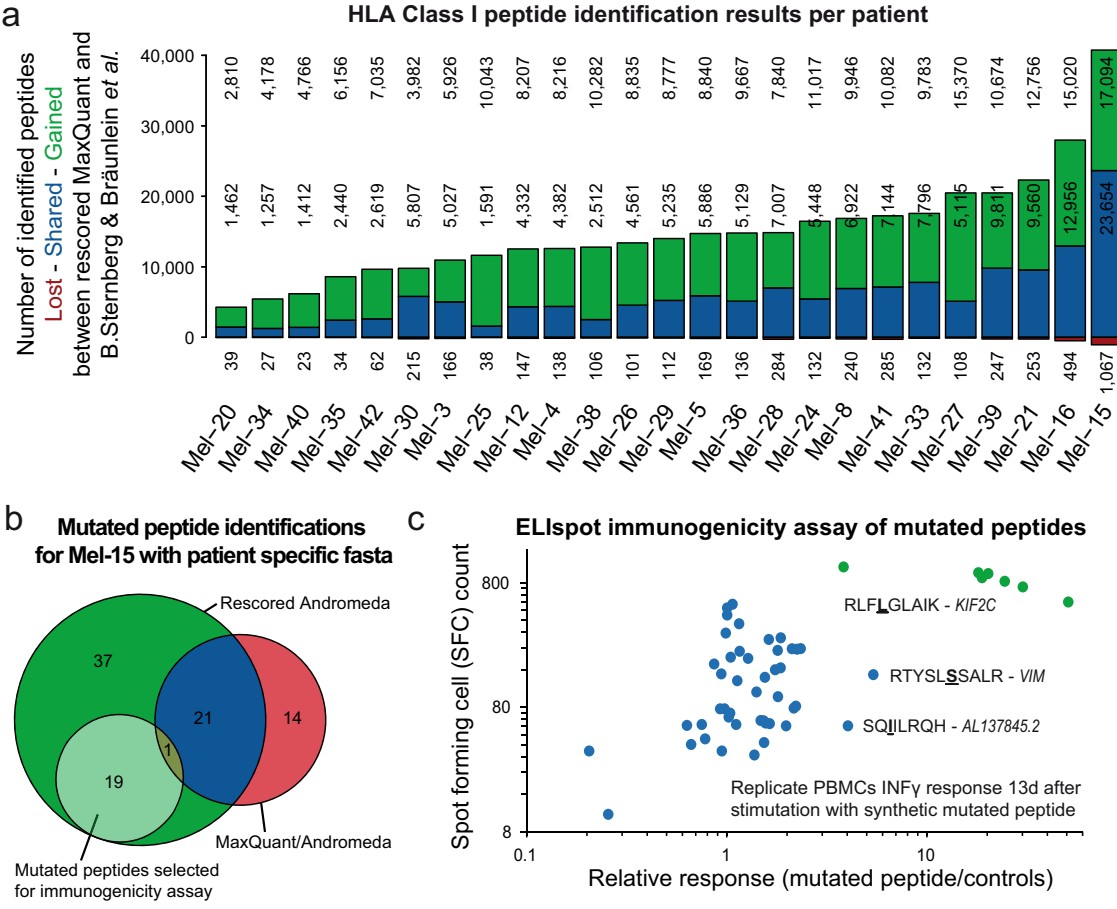

**Fig. 5 Prosit rescoring of 25 patient melanoma samples. a** Vennbars showing the number of peptides lost (red), shared (blue) and gained (green) by the MaxQuant rescoring pipeline compared to the results published by B.-Sternberg and Bräunlein et al.[11] for each patient. **b** Venn diagram comparing identified mutated neoepitopes by MaxQuant only (red) and by Prosit-based rescoring of MaxQuant only (dark green) and shared identifications (blue). Peptides selected for synthesis and further testing are depicted in light green. **c** Results of ELISpot immunogenicity assays probing synthetic mutated neoepitopes against Mel15 derived PBMCs. Readout was the number of spot forming cells probing interferon-gamma secretion. All batches and replicates were normalized to the respective positive controls. Between 3 and 6 replicates are plotted for each peptide. A *KIF2C* wildtype peptide, as well as unstimulated cells, served as controls. Assay data of replicates of the mutated *KIF2C* peptide RLFLGLAIK are highlighted in green. Mutations are shown as bold underlined amino acids. Raw and analysis data are available from the PRIDE repository with identifiers PXD021398, PXD004894, and Supplementary Data 7.

both the original study (8 neo-epitopes) as well as the MaxQuant results from this study (36 neo-epitopes). 14 peptides exclusively identified by MaxQuant were rejected by the rescoring pipeline due to their low spectral similarity (SA < 0.5) to the predictions (Supplementary Data 7).

From the list of additionally identified neo-epitopes, 20 were selected for immunogenicity testing in an accelerated co-cultured dendritic cell (acDC) assay using Mel15-derived PBMCs (Supplementary Data 7). Pure synthetic peptides were obtained and subjected to ELISpot analysis probing INF-γ secretion after stimulation of the PBMCs. Three peptide sequences triggered an immune response that substantially exceeded the response of unstimulated cells and canonical peptide controls (Fig. 5c). One particular peptide harboring a Proline to Leucine mutation in the *KIF2C* gene showed a very consistent immune response in all examined PBMCs batches (green dots). This mutation has been recently identified by a large-scale in silico approach based on MHC affinity predictions[25] of all peptides harboring mutations identified by RNASeq[6]. Our study presents solid experimental evidence (Supplementary Fig. S15e) that the mutated *KIF2C* gene is expressed and presented on the surface of tumor cells of the patient. In addition, we show the expression, presentation, and

immunogenicity of the mutated *VIM* gene[26]. Such direct evidence is important when it comes to assessing neo-epitopes for their therapeutic potential. As such, the Prosit rescoring workflow presented here constitutes a substantial improvement over purely computational approaches and can reduce the overall number of peptide candidates that need to be evaluated.

## Discussion

Artificial intelligence is revolutionizing many areas of research and is also making substantial contributions to mass spectrometry-based proteomics. As we and others have shown recently[7,27], it is possible to predict the retention times and tandem mass spectra of tryptic peptides with great precision which has led to substantial improvements in terms of the comprehensiveness and quality of proteomic experiments. Here, we demonstrated that the prediction of spectra using the deep learning architecture Prosit can be generalized to the simultaneous analysis of tryptic and non-tryptic peptides which greatly extends the range of experiments to which such predictions can be applied.

At the technical level, we note that while Prosit is currently limited to predicting b- and y-fragment ion intensities, it is robust against the presence of even a large number of neutral loss or

internal ion series. Prosit's ability to predict collision energies and to calibrate it to a particular dataset automatically removes the necessity to train dedicated models for individual projects, makes the analysis robust against variations in MS instrument parameters over time and, within reason, between mass spectrometers. In the future, Prosit may be further extended to include further fragment ion types as well as post-translational modifications. The need for the latter is particularly high for e. g. phosphorylated peptides because it is still difficult to assign the phosphorylation to a particular amino acid side chain. We also envisage Prosit to be implemented more broadly into peptide identification pipelines currently used in proteomics. This is expected to make results from different pipelines more comparable. In addition, it should also make the use of alternative proteases more successful than in the past.

The above examples demonstrate Prosit's practical utility for the analysis of HLA ligands—an application where spectral predictions turned out to be particularly successful. First, we show that Prosit more than doubles the number of HLA ligands that can be identified on the surface of human cells. Second, we provide strong evidence that the prevalence of proteasomal cis- or trans-splicing for the generation of HLA peptides is much lower than anticipated[24]. Third, we demonstrate that Prosit provides more comprehensive identification of neo-epitopes from patient tumors which, in turn, streamlines the subsequent testing and validation of these ligands as targets for immune-oncology. Many more biological and biomedical applications can be envisaged. To name a few, there is more and more evidence for the translation of small open reading frames (sORFs) which may be validated experimentally by Prosit re-scoring of MS data. Human body fluids contain rich peptidomes and glandular organs or cells secrete bioactive peptides all of which have not been comprehensively been mapped. To facilitate further research and applications, the rescoring toolchain is available online (https://www.proteomicsdb.org/prosit/).

## Methods

**Synthetic peptides**. Peptides selected for synthesis can be classified into four sets according to their origin. First, we generated a set termed HLA Class I that contained 168,688 peptides with 8–12 amino acids. The sequence list was compiled from entries in the Immune Epitope Database (IEDB, https://www.iedb.org[12]), where all human, MHC I binding peptides (status March 2017) were downloaded and the HLA large scale study from Bassani-Sternberg et al.[11]. Second, we generated a set termed HLA Class II that contained 73,464 peptides with 10-25 amino acids. Sequences were taken from the SysteMHC Atlas project (https://systemhcatlas.org), included were all sequences denoted as class II peptides (release April 2017)[13]. The datasets termed AspN and LysN consists of peptides derived from the digest with the respective proteases and contain 31,744 and 31,435 peptides with 7–25 amino acids. Peptide sequences were derived from the combined results of two large scale studies employing the respective protease for deep proteome studies of cell lines and tissues[28] and from an unpublished study (kindly provided by Josh Coon, Univ. of Wisconsin). The maximum number of peptides per protein was limited to the top 3 most frequently identified peptides per protein.

All sequences and protein mappings are available in Supplementary Data 1. Peptide Pool design, peptide synthesis, sample preparation, and LC-MS of synthetic peptides were previously described in Zolg and Wilhelm et al.[10] 2017, including the Supplementary Information. In brief, peptide pools for synthesis and measurement contained roughly 1000 peptides each. Near-isobaric peptides (±10 p. p.m.) were distributed across different pools of similar length to avoid ambiguous masses in pools wherever possible. All peptides were individually synthesized on cellulose membrane following the Fmoc-based solid phase synthesis strategy using a purpose-built peptide synthesizer[29]. The crude peptides were cleaved off the membrane in the predefined pools of 1000 peptides and dried. Dried peptide pools were initially solubilized in 100% dimethyl sulfoxide (DMSO) to a concentration of 10 pmol μl$^{-1}$ by vortexing for 30 min at room temperature. The pools were then diluted to 10% DMSO using 1% formic acid in high-performance liquid chromatography (HPLC)-grade water to a stock solution concentration of 1 pmol μl$^{-1}$ and stored at −20 °C until use.

**Data acquisition and database searching**. Ten microliter of the stock solution were transferred to a 96-well plate and spiked with two retention time standards (Pierce Retention Time Standard and PROCAL) at 100 fmol per injection[30]. An estimated amount of 200 or 500 fmol of every peptide in a pool was subjected to liquid chromatography using a Dionex 3000 HPLC system (Thermo Fisher Scientific) using in-house-packed C18 columns. The setup consisted of a 75 μm × 2 cm trap column packed with 5-μm particles of Reprosil Pur ODS-3 (Dr. Maisch) and a 75 μm × 40 cm analytical column packed with 3-μm particles of C18 Reprosil Gold 120 (Dr. Maisch). Peptides were loaded onto the trap column using 0.1% formic acid in water. We separated the peptides by using a linear gradient from 4% to 35% acetonitrile with 5% DMSO, 0.1% formic acid in water over 50 min followed by a washing step (60 min total method length) at a flow rate of 300 nl min$^{-1}$ and a column temperature of 50 °C[31]. The HPLC system was coupled online to an Orbitrap Fusion Lumos mass spectrometer (Thermo Fisher Scientific). Each peptide pool was first measured using a survey method consisting of an HCD (NCE, 28; Fourier transform mass spectrometry (FTMS)) and collision-induced dissociation (CID;NCE 35, ion trap mass spectrometry (ITMS)) fragmentation event. From these identifications, three further methods were created to target only full-length synthesis products (Xcalibur v4.2 and Orbitrap Fusion Lumos Tune v3.0 were used): (1) the 3×HCD method comprised HCD events at NCE 25, 30, 35 (all FTMS); (2) the 2xIT_2xHCD method comprised scans for CID NCE 35 ITMS, HCD NCE 28 ITMS, HCD NCE 20 FTMS, HCD NCE 23 FTMS; and (3) the ETD method comprised an ETD and FTMS scan (charge depended reaction time), electron-transfer/collision-induced dissociation (ETciD) NCE 35 FTMS and electron-transfer/HCD (EThcD) NCE 28 FTMS. For the peptide sets HLA Class I and HLA Class II mass ranges were specified each charge state to enhance coverage. Ranges were established by estimating the charge state by generating the 1% and 99%-percentile of the distribution of the synthesized peptide mass divided by the number of basic residues within the peptide. For HLA Class I the defined ranges were 145–1700 *m/z* for charge 3–6+; 280-1700 *m/z* for charge 2+ and 500–1700 *m/z* for charge 1+. For HLA Class II the defined ranges were defined as 230–2000 *m/z* for charge 3–6+; 490–1700 *m/z* for charge 2+ and 850–2000 *m/z* for charge 1+.

Acquired RAW data from synthetic peptides were analyzed using MaxQuant v.1.5.3.30[32] searching individual LC-MS runs against pool-specific databases[33]. If not mentioned otherwise, default parameters were used: carbamidomethylated cysteine was specified as fixed modification, methionine oxidation as variable modification. The first search tolerance was set to 20 p.p.m., main search tolerance to 4.5 p.p.m. and filtered for PSM and protein FDR of 1%. Visualization of MaxQuant result engine files was performed using custom R and python scripts.

In addition to the above-mentioned synthetic peptide pools, we also acquired data for the previously published proteoytpic and missing gene dataset[10] to assemble more training data for longer peptides. Pools that had an average peptide length above 25 were re-measured using the published settings, however without using an inclusion list for data acquisition. Using a regular DDA mode, allowed to record not only the full-length peptide products but also truncation products thereof. Accordingly, we performed the MaxQuant v.1.5.3.30 search using semi-tryptic digestion, allowing a free N-terminus.

### Deep learning using Prosit framework

*Fragmentation model and preparation.* Preparation of training data and deep learning of peptide fragment ion intensities was performed using the Prosit model architecture[7] using keras (2.1.1), tensorflow (1.4.0), numpy (1.14.5) and scipy (1.1.0). The peptide encoder consists of 3 layers: a bi-directional recurrent neural network (BDN) with gated recurrent memory units (GRU), a recurrerrent GRU layer and an attention layer with dropout. The recurrent layers use 512 memory cells each. The latent space is 512-dimensional. Precursor charge and NCE encoder is a single dense layer with the same output size as the peptide encoder. The latent peptide vector is decorated with the precursor charge and NCE vector by element-wise multiplication. A 1-layer length 29 BDN with GRUs, dropout and attention acts as decoder for fragment intensities. A keras model file was deposited in GitHub (www.github.com/kusterlab/prosit/) and zendoro with DOI zenodo.4721353[34]. In brief, the publicly available ProteomeTools data (PRIDE Dataset PXD004732, PXD010595), as well as the data presented in this study (PRIDE Dataset PXD021013), were utilized as training data. RAW data was searched using Max-Quant (version 1.5.3.30) using standard settings with 1% FDR filter at PSM, Protein or Site level. Unprocessed spectra for MaxQuant's rank 1 PSMs (from msms.txt) were extracted from the RAW files using Thermo Fisher's Raw-FileReader (http://planetorbitrap.com/rawfilereader) and b- and y-ions annotated for fragment charges 1 up to 3. Initial data included all PSMs for the same peptide and restricted peptides length to 7 to 30 amino acids length and precursor charge to <7 and Andromeda score to >40. NCE values of all runs were calibrated as described and spectra were transformed into a tensor format compatible with the machine learning models.

*Learning fragment ion intensities.* HCD Training data was split into three distinct sets with each peptide sequence only included in one of the three: Training (70%, ~593 k modified peptides, ~9.9 M PSMs), Test (20%, ~170 k modified peptides, ~2.8 M PSMs) and Holdout (10%, ~84 k modified peptides, ~ 1.4 M PSMs). CID Training data was split into three distinct sets with each peptide sequence only included in one of the three: Training (70%, ~500 k modified peptides, ~2.9 M PSMs), Test (20%, ~142k modified peptides, ~0.8 M PSMs) and Holdout (10%, ~71 k modified peptides, ~ 0.4 M PSMs). The model was trained and optimized on Training. Test was used to control for overfitting with early stopping. The Holdout dataset was used to

evaluate the model's generalization and potential biases. Normalized spectral contrast loss[7] was used as a loss function. We used the Adam optimizer with a cyclic learning rate (CLR) algorithm[35]. During training, the learning rate will cycle between a constant lower limit (0.0000001) and an upper limit (initially training started with 0.001, after restart for HCD 0.0001) which is continuously scaled by a factor of 0.95 every 8 epochs. Models were trained on a Nvidia TitanXp GPUs with 512 samples per batch. Models were trained on a Nvidia TitanXp GPUs for 195 epochs. During training, fragment ions out-of-range or out-of-charge were masked and not regarded in further analysis. For example, for a length 10 peptide y- and b-ions 10 to 29 as well as fragment ions with a charge higher than its precursor are masked. For training, the data was restricted to the top 3 highest scoring spectra for a peptide sequence, modification status, precursor charge, fragmentation method, fragmentation energy, and mass analyzer combination.

**General rescoring pipeline.** For the spectral intensity-based rescoring of search engine results [7]an unfiltered search result output including decoy PSMs (configuration and search engine for the individual datasets are detailed below) is used as input (minimal information content rawfile, scan number, modified sequence, and precursor charge state). Unprocessed MS2 spectra corresponding to the identifications were extracted from the RAW files using Thermo Fisher's RawFileReader (http://planetorbitrap.com/rawfilereader). The y and b ions of the extracted spectra were annotated at fragment charges 1 up to 3. Matching tolerances were 25 ppm for FTMS. We included all PSMs for the same peptide and restricted peptides length to 7–30 amino acids and precursor charge to <7, methionine oxidation as allowed variable modifications and cysteine carbamidomethylation as fixed modification. Annotation files were transformed to tensor format suitable for our machine learning models with a custom Python script. Ion intensities are continuous values and base-peak normalized. The best matching collision energy for prediction was automatically determined for every dataset by taking high scoring peptide identifications and comparing these to predicted spectra with NCE ranging from 20-40. Afterward, spectral comparison was performed for all spectra from the dataset using the determined best-matching CE and calculating previously described similarity measures (e.g., SA) between experimental and predicted fragment ion intensities using the annotated y and b ions. If not otherwise stated, we used the SVM Percolator 3.00 (https://percolator.ms)[36] with its standard settings for FDR calculation. Percolator was set to use the provided calculated spectral features to optimize for 1% FDR on PSM and 1% FDR on peptide level. Target-decoy-competition (-Y flag) was explicitly specified. Only spectral similarity features calculated in the rescoring process were used as features to achieve the best separation of target and decoys identifications, all search engine related scores were disregarded. The percolator input files with calculated feature sets, PSM and peptide level output tables files were used for visualization of the results using custom R scripts.

**Application of Prosit and rescoring to external datasets**

*Re-analysis of a large monoallelic HLA Class I cell line study.* For the analysis of the large monoallelic HLA Class I cell line study by Sarkizova and Klaeger et al.[16], raw data and the original Spectrum Mill (SM) HLA v2 search results were downloaded from MassIVE proteomics repository (identifier MSV000084172 and MSV000080527). The raw files corresponding to the three alleles B*07:01, B*35:01 and B*50:01 were excluded from the analysis. Remaining raw files were searched using MaxQuant 1.5.3.30 using the provided fasta file from the study. RAW files containing peptides that were not alkylated (noIAA) during sample processing were searched separate from RAW files containing alkylated peptides (IAA). The MaxQuant searches were performed using standard settings, each search was performed once with filtering for 1% FDR on PSM and once without (100%) filtering FDR on PSM level. Filters on protein level were not applied. The 100% FDR MaxQuant search results for IAA and noIAA were concatenated and rescored together. We further obtained unfiltered search engine results from SM HLA v3 (see below). The search result files from the improved SM HLA version (SM HLA v3) were rescored per allele. For this, the SM HLA output (ssv files) was converted into msms.txt-like files.SM provides the highest-ranking target and decoy match (sequence and score) per spectrum. During conversion, a spectrum is annotated as a target with the respective peptide sequence in case its score was higher or equal to the best decoy peptide match. Since SM annotates target and decoy cases that have an identical score as decoys, the Prosit handling may result in lower FDR estimates. Peptides which cannot be predicted with Prosit were counted as lost peptide for the analysis. These included peptides containing modifications for which Prosit was not trained: unmodified Cys, cysteinylated Cys, and acetylated peptide N-termini.

For comparison of the identification numbers (filtered for 1% FDR on PSM, and peptide level), either by allele or globally the search engine outputs of MaxQuant, the rescored MaxQuant data, the published SM HLA data, the improved SM HLA data, the rescored improved SM HLA data were visualized using a Vennbar plot. Vennbars are barplot representation of a venn diagrams, with proportional area corresponding to the overlap (blue area) and non-overlap (red and green) comparing two (data) sets. Sequence logos for all identified peptides in the original publication and peptides identified by either rescoring pipeline were visualized using the R library ggseqlogo[37]. Sequence logos for all alleles identified by respective search engines can be found in Supplementary Data 4.

The probability of a peptide originating from a given motif (referred to as emission probability) was estimated by the summation of the corresponding (peptide dependent) positional amino acids frequencies in the positional weight matrix. The positional weight matrix was generated using the only peptides previously published (SM HLA v2). In order to allow the comparison of emission probabilities across all investigated 8–11mers, only the top 5 most diverging positions in the positional weight matrix were used for summation. The ranking of the positions in each positional weight matrix was performed using the maximum observed frequency at every position in decreasing order, assuming that high values indicate high dependence on this position of a particular allele and peptide length. Emission probabilities were average across all investigated peptide length and reported for each allele separately (92 alleles).

*Reprocessing of HLA Class I monoallelic dataset with improved Spectrum Mill HLA v3 scoring.* Spectrum Mill (SM) v6.1 Pre-release with HLA v2 scoring was used in the large monoallelic HLA Class I cell line study[16]. Here, 92 out of the 95 alleles were reprocessed using SM v7.0 pre-release with the HLA v3 scoring update using the same protein sequence database searched for the previously published results. The primary enhancement to SM HLA v3 scoring is a revision to the noise level calculation performed for each spectrum prior to signal/noise based peak detection. This provides more sensitive peak detection in spectra of low abundance peptides with very little noise, and led not only to higher identification scores for low-signal spectra, but also allows more low-signal spectra to pass the sequence tag length based spectral quality threshold typically employed in SM prior to a database search. With the explicit goal of enabling re-scoring by Prosit of lower quality peptide spectrum matches with less-complete sequence coverage some SM thresholds were lowered. The sequence tag length spectral quality pre-filter was disabled, the threshold for minimum matched peak intensity was lowered from 30% to 10%, and the minimum identification score for result output was lowered 5 to 0. The re-processed searches also included a variable modification, acetylation of the protein N-termini, which was not previously available in SM for searches that are unconstrained by enzyme digestion specificity.

The following unchanged parameters from the previously published results[16] included: no-enzyme specificity; fixed modification: cysteinylation of cysteine; variable modifications: carbamidomethylation of cysteine, oxidation of methionine and pyroglutamic acid at peptide N-terminal glutamine; precursor mass tolerance of ±10 ppm; and a product mass tolerance of ±10 ppm. Variable modification of carbamidomethylation of cysteine was only used for HLA alleles that included an alkylation step (performed in 2017 or later). Peptide spectrum matches (PSMs) for individual spectra were automatically designated as confidently assigned using the SM autovalidation module to apply target-decoy-based FDR estimation at the PSM level of <1% FDR. Peptide autovalidation was done separately for each HLA allele with an auto thresholds strategy to optimize score and delta Rank1–Rank2 score thresholds separately for each precursor charge state (1 through 4) across all LC-MS/MS runs for an HLA allele. Score threshold determination also required that peptides had a minimum sequence length of 7, and PSMs had a minimum backbone cleavage score (BCS) of 5. BCS is a peptide sequence coverage metric and the BCS threshold enforces a uniformly higher minimum sequence coverage for each PSM, at least four or five residues of unambiguous sequence. The BCS metric serves to decrease false positives associated with spectra having fragmentation in a limited portion of the peptide that yield multiple ion types.

*Investigation of proteasomal splicing events using predicted fragment spectra.* As baseline for the investigation of proteasomal splicing events we downloaded the Mascot search result lists from a recent publication by Liepe et al.[21] from Mendeley (DOI: 10.17632/y2cvb5nvgn.1 [https://data.mendeley.com/datasets/y2cvb5nvgn/1]). Data used for analysis were the identification results from the HCD HCT116 and HCC1143 cell line MHC-I immunopeptidomes. The respective 7 raw files which were reanalyzed in the study, were obtained from the PRIDE repository from the original publication with identifier PXD000394[20].

For generation of a percolator model, the raw files were searched with MaxQuant 1.5.3.30 against a human Swiss-Prot protein sequence database including annotated isoforms (downloaded 2 July 2016; 42,164 protein sequences) using default settings, but no enzyme specificity defined and no FDR filtering on any level. Spectra for resulting target and decoy PSMs were extracted from the raw file and rescored as described above with the difference that Percolator v3.05[36] was employed for FDR calculation taking into account only features that are calculated during the rescoring and neglecting all search engine derived scores. The percolator input files with calculated feature sets, PSM and peptide level output tables files were used for visualization of the results using custom R scripts.

As the obtained Mascot results of the alleged spliced and non-spliced peptide identifications did not contain comprehensive target and decoy information and were filtered according to the Mascot ion score, rescoring of the peptide identifications had to be performed with a pre-trained percolator model. Hence, the general rescoring pipeline was executed as described above, however only the PSMs recorded in the Mascot output were rescored and handed to Percolator for FDR estimation, using the Percolator model pre-trained on the MaxQuant results. This had the favorable effect that the results from the different search engines were in the same confidence space and could directly be compared. Again, the percolator

input files with calculated feature sets, PSM and peptide level output tables files were used for visualization of the results using custom R scripts.

For MSFragger analysis, MSFragger v3.0[38] was used in conjunction with FragPipe v12 GUI and the Swiss-Prot protein sequence database including annotated isoforms (downloaded 2 July 2016; 42,164 protein sequences) was used. Default parameters for a closed search were employed, with a defined precursor tolerance of 20 ppm, mass recalibration switched off and no enzyme specificity for database digest. Raw files were searched without decoy database, no PSM or peptide FDR estimation was performed. Resulting PSM lists were submitted to the rescoring pipeline as described above and handed to Percolator for FDR estimation, using the Percolator model pre-trained on the MaxQuant results. Again, the percolator input files with calculated feature sets, PSM and peptide level output tables files were used for visualization of the results using custom R scripts.

For all proposed spliced peptide spectra from the Liepe et al.[21] dataset, we generated mirror plots comparing the raw spectrum with the predicted spectrum of the alleged peptide identification by Mascot, MaxQuant, and MSFragger. Proposed spliced peptides that did get rejected based on the results of one of the other search engines are marked with rejected in the  Supplemental File 2.

*Rescoring of melanoma patient HLA dataset.* For the reprocessing of a large clinical cohort of melanoma patients, submitted RAW and search data from the original study were obtained from the PRIDE repository with the identifier PXD004894[11]. To apply the rescoring toolchain, the RAW files from the study were reprocessed (HLA Class I and Class II RAW files separately) using MaxQuant 1.5.3.30 and the human fasta file provided from the original study (HUMAN_2014 fasta containing Swissprot and Trembl identifiers and a total of 85.919 entries). Standard MaxQuant settings were used for processing but no enzyme specificity and no FDR filtering on PSM or protein level was applied. The results of this search were then rescored as described above. Results of the rescoring pipeline were compared to the original study data and visualized using the described Vennbar plot.

To assess the number of mutated peptides in the patient Mel15, RAW files belonging to patient Mel15 were processed using MaxQuant 1.6.1.0 with a concatenated protein sequence database established from the ensemble release 92 database (107,844 protein sequences) and Mel15 specific mutated sequences (126,906 protein sequences). Mutant protein sequences were derived from the analysis of patient-specific whole-exome sequencing (WES) and RNA-Seq data. Mutation calling was performed using MuTect2 4.1.0.0[39] for WES and Strelka2 2.9.10[40] for RNA-Seq data. All coding and non-coding transcripts containing one or more non-synonymous somatic mutations were then translated into corresponding amino acid sequences. Two MaxQuant searches were performed: One using 1%/100% FDR on PSM and protein level for comparison and one using 100%/100% PSM and protein level FDR for rescoring. We estimated the resulting peptide FDR from the MaxQuant 1%/100% to approximately 3% FDR and adjusted the rescoring peptide FDR accordingly. Candidate peptides that exclusively matched into the database containing the patient-specific sequencing derived proteins with mutations were flagged as candidates. We further manually annotated and filtered the identified peptides to either contain a mutation within the peptide or selected peptides where a frameshift upstream mutation would cause the expression of such a peptide. Mutated peptide sequences were submitted to NetMHC4.0[41] server to retrieve binding affinity predictions for the Mel15 alleles HLA-A*03:01, HLA-A*68:01, HLA-B*27:05, and HLA-B*35:03 analogous to a previous publication[11] using standard settings. Resulting classification als weak binder or strong binder are available in Supplementary Data 7. Similarly, mutated peptide sequences were submitted to the HLAthena[16] webservice [http://hlathena.tools] for binding prediction to the above-stated alleles using standard settings and a rank based threshold of 0.1. Resulting predicted binding are available in Supplementary Data 7.

*Immunogenicity assessment of identified peptide ligands.* Informed consent of all healthy and diseased participants was obtained following requirements of the institutional review board (Application 193/17S; Ethics Commission, Faculty of Medicine, Technical University of Munich, Germany). Recall antigen-experienced T cell-responses to 20 selected peptides were investigated as previously described with slight modifications[11,42]. In brief, between 0.3 and 0.7 Mio PBMCs per well derived from six different blood samples from patient Mel15 were used for in vitro screening. For peptide stimulation, 1 μM of each synthetic peptide (>90% purity, ordered from DGPeptidesCo Ltd.) was added to the culture along with 0.5 ng/ml Interleukin (IL)−7 (Peprotech), 50 ng/ml Tumor necrosis factor (TNF)-α (Peprotech) and 10 ng/ml IL-1β (Peprotech). As antigen-presenting target cells for the second stimulation on day 13, a Lymphoblastoid cell line (LCL) derived from the same patient was used. The target cells were either pulsed with the selected mutated peptide, an irrelevant peptide (KIF2C wildtype) or without any peptide prior to co-culture with the T cells. The co-cultures were performed with an effecter-to-target ratio of 1:2 using 10,000 target and 20,000 pre-stimulated T cells per well.

Reactivity of T-cells to the synthetic peptide ligands was assessed by specific Interferon (IFN)-γ release by ELISpot assay after 1 day and 13 days. ELISpot plates (MAHAS4510) were coated with an IFN-γ capture antibody (1-D1K, Mabtech), development was performed with an IFN-γ-detection antibody (7-B6-1-biotin, Mabtech) and Streptavidin-HRP (Mabtech). ELISpot plates were read out on an ImmunoSpot S6 Ultra-V Analyzer using Immunospot software 5.4.0.1

(CTL-Europe). The average normalized ELISpot intensity at day 13 to the average of the controls was used for data visualisation.

**Reporting summary**. Further information on research design is available in the Nature Research Reporting Summary linked to this article.

## Data availability
Reference spectra for synthetic peptide originating from the proteases LysN and AspN are available at https://www.proteomicsdb.org, and updates to the resource are available at https://www.proteometools.org. Updated models of Prosit (Prosit_2020_intensity_HCD and Prosit_2020_intensity_CID) and the presented rescoring functionality are freely available through the web interface at https://www.proteomicsdb.org/prosit. Trained model files (https://figshare.com/articles/dataset/Prosit_Non_tryptic_-_Model_-_Fragmentation/12936947) and training data (https://figshare.com/articles/dataset/ProteomeTools_non_tryptic_-_Prosit_fragmentation_-_Data/12937092) are available on figshare. The mass spectrometric raw and search data of the synthetic ProteomeTools peptides have been deposited with the ProteomeXchange Consortium via the PRIDE repository with the dataset identifier PXD021013. The search data including intermediate results underlying the presented analysis have been deposited under the dataset identifier PXD021398.

## Code availability
Source code and scripts are available on GitHub at https://github.com/kusterlab/prosit/ and zenodo with DOI zenodo.472135[34]. Custom analysis scripts are available upon request. Prosit rescoring can be performed online at ProteomicsDB [https://www.proteomicsdb.org/prosit/].

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

## Acknowledgements

The authors thank the members of the Kuster lab for fruitful discussions and the Coon lab for providing access to a list of identified peptides from an (at the time of writing) unpublished study allowing the selection of proteotypic AspN and LysN derived peptides. This work was in part funded by the German Federal Ministry of Education and Research (BMBF; grant No 031L0008A and 031L0168), European Union's Horizon 2020 Program under Grant Agreement 823839 (H2020-INFRAIA-2018-1; EPIC-XS), EIT Health through the grant No. 19638 (EIT Health is supported by the EIT, a body of the European Union) and DFG (SFB824/C10). The Titan XP used in this research were donated by the NVIDIA corporation. The IBM infrastructure hosting ProteomicsDB and Prosit are operated and maintained by the UCC at the TUM. This work was supported in part by grants from the National Cancer Institute (NCI) Clinical Proteomic Tumor Analysis Consortium grants NIH/NCI U24-CA210986 and NIH/NCI U01 CA214125 (to S.A.C.). S.K. is a Cancer Research Institute/Hearst Foundation fellow.

## Author contributions

A.H., U.R., M.W., and B.K. conceived the study. M.W., D.Z., C.S.W., E.B., Ph.S., R.R., S.C., C.R.K., A.M.K., and B.K. designed experiments. M.W., D.Z., M.G., S.G., T.S., C.S. W., and Ph.S. performed experiments. M.W., D.Z., M.G., S.G., T.S., C.S.W., E.B., Ph.S., N.A.K., S.K., B.D., S.C., K.R.C., and A.M.K. analyzed data. T.S., Pa.S., and L.L. extended the web resources. K.S., J.Z., T.K., U.R., and H.W. optimized, performed, and oversaw peptide synthesis. M.W., D.Z., K.R.C. and B.K. wrote the manuscript.

## Funding

## Competing interests

D.P.Z., S.G., T.S., M.W., and B.K. are founders and shareholders of MSAID GmbH. M.W. and B.K. are also founders and shareholders of OmicScouts GmbH, they have no operational role in both companies. K.S., J.Z., T.K., and U.R. are employees of JPT Peptide Technologies. B.D. and A.H. are employees of Thermo Fisher Scientific. During the study, S.G. was an employee of SAP SE. S.A.C. is a member of the scientific advisory boards of Kymera, PTM BioLabs, Seer and BioAnalytix and a scientific advisor to Pfizer and Biogen.
