## [Peer Review File · Nature Communications]

Reviewers' Comments:

Reviewer #1:

Remarks to the Author:

Based on their previously developed mass spectrometry identification tool Prosit using a deep neural network computing framework, this paper proves that Prosit can be extended to analyze trypsin and non-trypsin peptides at the same time, further expanding the range of mass spectrum experimental data which could be identified. The precision and robustness of Prosit's extended non-trypsin peptide mass spectrometry data analysis capabilities are impressive.

This work also produced 9 million HCD mass spectra data of massive amounts of non-tryptic peptides used for training. These data are very valuable and can help researchers in the field to carry out related studies of calculation methods and algorithms research. This technical method is of great significance to the research of immuno-proteomics, and the developed non-trypsin peptide identification technology has laid the foundation of immuno-proteomics research. The work of this paper is very important, the methods and data are extremely valuable, and the results are impressive. It is recommended to accept for publishing. Some small comments:

1. Compared with the previous work of Prosit published by the team on Nature Method, what are the important technical improvements and innovations in this paper?
2. Deep neural network has strong learning ability, but consumes a lot of computing resources and sometimes has serious over-fitting problems. Have you compared the advantages and disadvantages of the extended Prosit and other alternative calculation methods? What is the result?
3. The paper mentions that Prosit's technical framework can also be extended to the identification of post-translational modification mass spectrometry data. What are the key breakthroughs in this possible technological expansion? Are there any general key technological breakthroughs in the step-by-step expansion technology? For example, a breakthrough in unsupervised learning technology?

Reviewer #2:

Remarks to the Author:

Accurate prediction of relative ion intensities of tandem mass spectra is a long sought goal in mass spectrometry based proteomics. The authors have previously released a deep-learning based tool Prosit. In this study they have updated this resource by training the model with new mass spec experimental data of ~300k non-tryptic peptides. They show that the prediction of non-tryptic peptide fragmentation spectra improved. The new model was then applied toward identifying and rescoring MHC-bound immunopeptides which improved identification rates. Overall this study looks promising and should improve upon a useful resource. I have the following comments:

1. The authors first compared the performance of 2020 Prosit model against the 2019 model that was trained on tryptic peptide.
 - * The improvement of prediction for non-tryptic peptide looks surprisingly modest (Figure 1b, also comparing with all peptides in Supp Fig S7), which is surprising given the potential fragmentation differences between peptides with tryptic and non-tryptic ends. For instance, spectral angle 0.915 vs. 0.848 for MHC I peptides. Interestingly it appears the median spectral angle in the 2019 model was lowered by a long tail of poor predictions. Did the authors investigate what those peptides poorly predicted by the 2019 model (trained using tryptic peptides) were (e.g., particular sequence features? singly charged peptides)?
 - * On a related note, a general question about the magnitude of improvement in the Prosit 2020 model is whether the comparison metrics are sensitive to different measures of spectral similarity, e.g., spectral angle encodes the entire spectrum and all peaks as a multidimensional vector and so would be more sensitive to minor differences, but most proteomics search engines don't score spectral similarity with spectral angle. What is the implication here and what would be the result if spectra were compared using other metrics, e.g., skyline dotp or sequest-style xcorr?

* Does the Prosit 2020 model also improve on predictions of tryptic peptides with internal basic sites (histidine or miscleavages) or chymotryptic peptide?

2. The authors then reanalyzed the MHC-bound peptide data from Sarkizova et al. and found that rescoring the candidate PSMs using the 2020 Prosit-predicted spectral intensity improved identification (~50%) (main text line 181 and Supp Fig 11).

* Given the major novelty of the current study, is the improvement of the Prosit model through training of new non-tryptic peptides, a fairer comparison would be to compare the 2020 Prosit-based rescoring with similar rescoring using the 2019 Prosit model or other approaches that predict fragment ion intensities (e.g., MS2PIP, pDeep).

3. The authors also re-analyzed data related to proteasome-spliced peptides and found a large proportion of false positives (87%; line 253). This promises to be a useful result and adds to the ongoing debate about the extent of these events.

* The Liepe et al. Science 2016 study attempted to validate 98 spliced peptides against synthetic peptides. Are any of those peptides captured in this reanalysis and do they show better Prosit prediction than other, un-validated peptides?

* In general I believe this section should be more precisely worded, as well as the Discussion line 320 ("concept of proteasomal [splicing] ...is not well founded"). The workflow here arrived at the same conclusion of some previous analyses that many of the putative proteasome-spliced peptides have low search scores. This is consistent with a relative rarity of proteasome spliced peptides, but nevertheless the workflow reconfirmed ~3% of the HLA identified peptides to be spliced, some of which may be important for the search of cell-specific epitopes.

* Are there scoring differences between the mechanistically more intuitive cis-splicing vs. trans-splicing?

* It would be useful to see if the false positive spliced peptides also have different experimental v.s predicted retention time (e.g., in Rolfs et. al. Sci Immunology 2019)

* Line 248: "The Percolator score difference between the proposed spliced (Score = 2.9) and canonical peptide (Score = 2.5) is small." What are the Percolator q value and PEP for the spliced vs. non-spliced peptides, since Percolator already takes into account the scoring difference between the first and the second hit?

4. The authors reanalyzed the melanoma MHC1/MHC2 peptides in Bassani-Sternberg et al. Nat Comm 2016 and found additional peptides including neo-epitopes.

* There should be a more comprehensive audit on the nature of the gained identification via Prosit 2020 rescoring. For the spectra responsible for the gained identification, what sequences were they identified to in the original pipeline? E.g., were the best PSM identified to the same sequence but not passing FDR cutoff without rescoring? How close were the rescored vs. original PSM in terms of scoring? Regardless of whether the PSM newly passed Percolator PEP/FDR cutoff, I believe there should be an attempt to quantify or at least visualize the gained explanatory power on a per-spectrum level, similar to what was done for spliced peptides in Supplementary File 2.

* Line 281, it is not clear why the Maxquant reanalyses here were able to reidentify substantially more (36 vs. 8) melanoma immunopeptides over the original Bassani-Sternberg et al. study, since Maxquant was also used in that study.

5. Overall it would be useful to see a more in-depth analysis of where the improvement in the Prosit 2020 prediction comes from and what features might be important for accurate MS2 prediction, e.g., what ions are given more intensities in the 2020 models the case of an internal basic residue. It is not surprising per se for a deep learning model to perform better when being fed substantially more data, but there is a great opportunity here to further dissect the black box of machine learning predictions.

Reviewer #3:

Remarks to the Author:

Wilhelm et al. have submitted a manuscript on improved prediction of MS/MS spectra of HLA ligands and non-tryptic peptides using previously established deep learning framework Prosit. The authors performed multi-modal LC-MS/MS measurements of over 300,000 peptides representing HLA class I & II ligands and products of the proteases AspN and LysN, and used the resulting nine million MS/MS spectra to re-train the Prosit platform. They report significant improvements in prediction of MS/MS spectral features by the new version of Prosit, especially for singly charged, non-tryptic peptides, but also for higher charged peptides and other commonly used types of fragmentation, such as CID. As a result, Prosit is now significantly improved and well suited for immunopeptidomic applications across several MS platforms. The Authors demonstrate improvement in the number of HLA peptide identifications by over 25% after re-processing the MS/MS spectra obtained in the benchmark dataset of Sarkizova et al. They further apply the re-trained Prosit platform to demonstrate that about 90% of the "spliced" peptides previously reported by Liepe et al. are either wrong, or can be explained by canonical proteasomal processing. Finally, they apply Prosit to another landmark study of Bassani-Sternberg et al. that analyzed HLA-ligandomes in 25 melanoma patients; importantly, the Authors detect 7-fold more HLA class I peptides in this dataset and successfully follow up several of the novel HLA epitopes.

In summary, this is a carefully executed large-scale MS study that significantly improves an established platform for prediction of MS/MS spectral features (Prosit). The re-trained platform has a potential to revolutionize immunopeptidomics, accelerate development of anti-cancer vaccines and personalized medicine in general. Moreover, the study addresses and challenges the controversial concept of proteasomal peptide splicing. This is the only part of the manuscript where I missed a more thorough analysis of the available datasets (more MS files), as well as more detailed treatment of the "surviving" spliced HLA peptides; e.g. it would be good to know what kind of scores/intensities they have because even this low number may point to the existence of HLA peptide splicing at very low levels or in a small population of cells. Although this part will surely incite a discussion, I see this as a minor point and I fully recommend publication of the manuscript in Nature Communications.

Reviewer #4:

Remarks to the Author:

The authors synthesized and analyzed the LC-MS data of more than 300,000 peptides of HLA class I, HLA class II, and other non-tryptic peptides. The data was then used to train a deep learning model, Prosit, to predict MS/MS spectra of HLA peptides. The spectra prediction model improved the identification of HLA peptides by database search engines such as MaxQuant, MSFragger, and subsequently might lead to the identification of more neo-epitopes for cancer immunotherapy.

Detailed comments:

The title is not clear. The title should briefly state the research problem/results of the paper. When reading the current title, I don't understand, for example, what "one mass spectrum at a time" refers to. In addition, both "deep learning" and "immunopeptidomics" are very broad topics, so it is not clear what specifically this paper is trying to do. I think the main contribution of this paper is the synthetic HLA peptides, so it should be reflected in the title.

Lines 55-56: ">300,000 peptides by multi-modal LC-MS/MS within the ProteomeTools project representing HLA class I & II ligands and products of the proteases AspN and LysN"

It's better to clearly state the number of peptides for each class as in lines 94-95.

Line 59: "the identification of HLA peptides can be greatly improved"

Please state specifically how much % the improvement was and for which database search engines.

Line 59: "proteasomal HLA peptide splicing may not occur to any noticeable extent"

This is not clear, please elaborate.

Line 102, Figure 1c: the column and the row corresponding to amino acid Cysteine (C) seem to have very low number of peptides, maybe the authors could provide more explanation why this is the case.

Figure 1e: Could we have the distribution of Andromeda score for tryptic peptides for comparison? And the distribution of Andromeda score for endogenous HLA-I and HLA-II peptides (where data is available, for example, reference 11)

Lines 138-141: "Combining the 9 million HCD spectra of non-tryptic peptides collected in this study with the 21 million spectra previously published tryptic peptide enabled the training (70% of all spectra), testing (20%) and validation (10%, referred to as holdout set) of a single high-performant Prosit model for both types of peptides (Figure 2a, Supplementary Figure S6a, Methods)"

The authors emphasized throughout the manuscript about "a single Prosit model" for both tryptic and non-tryptic peptides. While using a single model maybe more convenient, one could argue that specific models for each type of peptides would provide more accurate predictions. Could we have a comparison between a single model versus specific models? Given the amount of data available in this paper, I believe it is sufficient to train specific models. As the authors have mentioned, these two types of peptides are very different in fragmentation characteristics, so specific models are desirable. My other concern with the single model is how to deal with the class imbalance, e.g., the amount of tryptic spectra is more than double that of non-tryptic spectra, so the single model is definitely bias towards tryptic spectra after training. Last but not least, HLA peptides are very diverse and depend on a large number of different HLA alleles. In the current manuscript, there is no analysis/discussion about HLA alleles of the training data. Furthermore, it would be interesting to explore if one could train a model specific to each HLA allele, as we are moving towards the direction of personalized immunotherapy.

Regarding the training-validation-testing 70-20-10 partition, from my understanding, this partition is based on the spectra. However, different spectra may share the same peptide sequence, since we have near 30 million spectra but less than 1 million peptides. So it is possible that the training, validation, and testing sets may share common peptides. Deep learning models like Prosit may simply memorize those common peptide sequences from training and apply to testing, so the accuracy may not be accurate because we need to test on peptide sequences that the model has not seen before. Lines 159-161, Figure 2e: Figure 2e has little meaning because it is certainly that training accuracy is always better than testing accuracy. One can compare different models on the same dataset, or different datasets on the same model (like Figure 2b), but training versus testing of the same dataset does not show anything interesting.

Lines 172-177, Figure 3a: It would be interesting to show how much improvement in spectra prediction would lead to how much improvement in identification, especially for HLA peptides. For instance, is SA=0.90 is good enough for identification, or is it worth to try to reach SA=0.99 with more data or larger models?

Lines 191-194: I don't quite understand. The authors compared MaxQuant standalone versus MaxQuant+Prosit. So how was the loss of 15,000 peptides related to SM HLA v2? It is better to show the score histograms of MaxQuant results versus MaxQuant+Prosit and highlight the common, extra, and loss peptides to see how confident those identifications are.

I think it is better to perform this performance comparison on the synthetic HLA peptides of this manuscript. Because otherwise, Prosit was already published in a previous paper and so was the dataset of 95 monoallelic cell lines, so there is little novelty here. I believe the main contribution of this manuscript is the dataset of synthetic HLA peptides and the analysis should focus on this dataset, rather than on Prosit and other things (which could be placed in the supplementary).

Lines 226-230: I'm not sure whether this comparison is fair. The Prosit model was trained on non-spliced peptides, so one could expect its performance on spliced peptides was not as good as on non-spliced peptides. Particularly, Prosit is a deep learning sequential model, so it may memorize non-spliced peptide sequences but it has not seen any spliced peptide sequences during training. I would suggest the authors to re-train Prosit model on a fraction of spliced PSMs and test it on the remaining spliced PSMs.

In general, I don't think a universal Prosit model should be applied anywhere to any dataset, as I have mentioned above for tryptic and non-tryptic peptides. For example, the performance of non-spliced peptides in Figure 4a (SA=0.87) is worse than the results in Figure 2b (SA=0.92), despite that they are both HLA non-spliced peptides. So the model performance depends on the training data, and re-training is desirable when dealing with a new type of data such as spliced peptides. Apparently the

binding motifs of spliced peptides are different from those of non-spliced peptides, and so are their amino acid compositions and hence fragmentation characteristics. Subsequently, I think it is not appropriate to use a Prosit model that was trained on non-spliced peptides to re-score database search results of spliced peptides from MaxQuant and/or MSFragger. So I don't think the authors' analysis and argument against proteasomal splicing are convincing.

Lines 273-275, Figure 5a: Please show the score distribution of newly identified PSMs superimposed on that of existing PSMs to see how confident the new identifications are.

Lines 279-284: I believe the authors of reference 11 (Bassani-Sternberg et al.) used MaxQuant in their proteogenomic analysis and identified only 8 neo-epitopes. Yet the current manuscript also used MaxQuant and identified 36 neo-epitopes. Please explain why there is such a big difference, did you use the same FDR threshold. Please also show the identification scores, FDR, binding affinity, and immunogenicity of those neo-epitopes (similar to Table 1 of reference 11)

Lines 288-291: How is the immune response of the new neo-epitopes compared to those in the original study (reference 11)?

The peptide "RTYSLSSALR" overlapped with the peptides reported recently by Tran et al. on the same patient Mel-15 dataset (<https://www.nature.com/articles/s42256-020-00260-4>), where it was suggested to belong to HLA-II. Apparently the binding motif, 2nd and 9th amino acids (T and R, respectively), do not fit into any HLA-I alleles of patient Mel-15 (Figure 1b of reference 11). In contrast, Tran et al. reported multiple HLA-II peptides encompassing this mutation.

I found the number of 78 neo-epitopes of patient Mel-15 quite mind-boggling. Figure 5a shows the number of peptides of patient Mel-15 increased by 17,094 on top of existing 23,654, i.e. about 72% improvement. Yet the number of neo-epitopes increased by 10 folds (78 versus 8). So please list all the neo-epitopes, their identification scores and PSMs, etc.

Reviewer #1 (Expertise: Proteomics and Mass Spectrometry):

Based on their previously developed mass spectrometry identification tool Prosit using a deep neural network computing framework, this paper proves that Prosit can be extended to analyze trypsin and non-trypsin peptides at the same time, further expanding the range of mass spectrum experimental data which could be identified. The precision and robustness of Prosit's extended non-trypsin peptide mass spectrometry data analysis capabilities are impressive.

The authors are happy to read that the reviewer appreciates the value of the new Prosit models.

This work also produced 9 million HCD mass spectra data of massive amounts of non-tryptic peptides used for training. These data are very valuable and can help researchers in the field to carry out related studies of calculation methods and algorithms research. This technical method is of great significance to the research of immuno-proteomics, and the developed non-trypsin peptide identification technology has laid the foundation of immuno-proteomics research. The work of this paper is very important, the methods and data are extremely valuable, and the results are impressive. It is recommended to accept for publishing. Some small comments:

The authors are also grateful for these comments. We share the view that the MS data provided alongside Prosit will be valuable to others who may use the data for creating further and improved software tools.

1. Compared with the previous work of Prosit published by the team on Nature Method, what are the important technical improvements and innovations in this paper?

The three main points of technical improvement and innovation are:

- a) The fact that predictions are now available for any peptide of length 7-30 amino acids regardless whether it is tryptic or not. HLA peptides are just one example for such peptides and for which the new version is particularly useful/important.
- b) The current work also provides a model for (resonance type) CID fragmentation (in an ion trap). This fragmentation technique is used a lot in proteomics and the fragmentation characteristics of peptides in CID are quite different to those of (beam type) HCD (in a multipole). Therefore, the current work goes far beyond the 2019 version of Prosit that was confined to the analysis of HCD data.

- c) We now show that the rescoring approach can be applied to peptide identification results of different search engines. While this alone might not be too surprising, we show that it is now possible to compare results from different search engines directly.

In addition, we have observed that Prosit is used very widely by the proteomics research community. Over the past year, our web service provided predicted spectra for more than 2 billion (!) peptides/precursors. We expect that the extended capabilities provided with the current version, will increase the use of this service even further.

2. Deep neural network has strong learning ability, but consumes a lot of computing resources and sometimes has serious over-fitting problems. Have you compared the advantages and disadvantages of the extended Prosit and other alternative calculation methods? What is the result?

Regarding computing resources, this is not a substantial issue. First, because the authors provide a service to the community via ProteomicsDB and many researchers are using this facility to re-score their own data (see above). Second, a stand-alone version of Prosit predicts spectra at a rate of $\sim 30,000$ /s spectra per second on a desktop PC (with GPU) which is sufficient for many purposes. Third, GPU computing is becoming more readily available which alleviates the need for access to specialized computing facilities.

Regarding overfitting, we control this very carefully by (a) separating training data from testing and validation (holdout) data (see Methods and comment to reviewer #4), (b) using a rather high dropout during training, and (c) early stopping of the training step. We show in Supplementary Figure S10a-c (for HCD) and Figure 2e (for CID) that there is indication of no substantial overfitting. This can be judged by comparing the spectral angle distributions between the training and holdout sets. We had already controlled overfitting in this way for the previous Prosit model.

Regarding other calculation methods: we had already shown comparisons of Prosit vs MS2PIP (for HCD) in our 2019 paper in Nature Methods. For the purpose of answering the reviewers question, we have compared Prosit CID with that of MS2PIP. MS2PIP reports a Pearson correlation of 0.90 (for singly charged fragments) and 0.81 (for doubly charged fragments). Prosit CID achieves a Pearson correlation of ~ 0.98 (singly, doubly and triply charged fragments). That said, the authors anticipate that the large amount of MS data supplied along with our manuscript will be used by others to extend the aforementioned other spectrum prediction tools to non-tryptic peptides.

The major strength of deep learning in general, and the layer architecture used for Prosit in particular, is its ability to generalize. We have observed this already in the 2019 publication, where Prosit showed decent prediction accuracy for e.g. chymotryptic peptides despite the fact that it was not trained for such cases. This was not the case for e.g. MS2PIP which uses classical machine learning.

3. The paper mentions that Prosit's technical framework can also be extended to the identification of post-translational modification mass spectrometry data. What are the key breakthroughs in this possible technological expansion? Are there any general key technological breakthroughs in the step-by-step expansion technology? For example, a breakthrough in unsupervised learning technology?

The authors are not sure they fully understand the questions. Right now, Prosit can only deal with oxidized methionine as a PTM because it has not been trained yet to recognize/learn the fragmentation characteristics of e. g. phosphorylated peptides. However, the authors plan to do so based on a large number of synthetic PTM peptides that have been synthesized as part of the ProteomeTools project (doi: 10.1038/nmeth.4153). Based on ongoing but preliminary work, we are confident that the current Prosit framework will be able to learn and predict fragment intensities of modified peptides too. However, this requires further work and will, therefore, be reported at another time. In our view, one of the main strengths of the 2020 Prosit model is that it works well as a single model for tryptic and non-tryptic peptides rather than requiring separate models. We also think that a single model can be generated to include e.g. phosphorylation, but again, additional work is required before we can release such a model. Please also see our comments to reviewer #4 below.

Reviewer #2 (Expertise: Mass spectrometry and neopeptidomics):

Accurate prediction of relative ion intensities of tandem mass spectra is a long sought goal in mass spectrometry based proteomics. The authors have previously released a deep-learning based tool Prosit. In this study they have updated this resource by training the model with new mass spec experimental data of ~300k non-tryptic peptides. They show that the prediction of non-tryptic peptide fragmentation spectra improved. The new model was then applied toward identifying and rescoring MHC-bound immunopeptides which improved identification rates. Overall this study looks promising and should improve upon a useful resource. I have the following comments:

1. The authors first compared the performance of 2020 Prosit model against the 2019 model that was trained on tryptic peptide.

* The improvement of prediction for non-tryptic peptide looks surprisingly modest (Figure 1b, also comparing with all peptides in Supp Fig S7), which is surprising given the potential fragmentation differences between peptides with tryptic and non-tryptic ends. For instance, spectral angle 0.915 vs. 0.848 for MHC I peptides. Interestingly it appears the median spectral angle in the 2019 model was lowered by a long tail of poor predictions. Did the authors investigate what those peptides poorly predicted by the 2019 model (trained using tryptic peptides) were (e.g., particular sequence features? singly charged peptides)?

We had already noted in the 2019 publication that Prosit trained on tryptic peptides worked surprisingly well (albeit not nearly as well) on non-tryptic peptides. At the time, we speculated that the neural network underlying the Prosit 2019 model had learned fragmentation rules that go beyond the knowledge that the field empirically accumulated about the MS/MS behavior of peptides that terminate in K or R residues. This can in part be explained by the fact that the original training data for the 2019 Prosit model contained 11,075 sequences (2.5% out of a total of 426,564 peptide sequences with Andromeda Score >70) which represent the C-terminus of proteins and which are “non-tryptic” because they do not end with a K or R residue. Given this rather modest number, we could not be more affirmative about this observation at the time. Indeed, this was one of the specific reasons why we subsequently synthesized, measured and trained on a large number of non-tryptic peptides to address this shortcoming. Just for clarification, we note that the 2020 Prosit model was generated by training on tryptic and non-tryptic peptides at the same time. As mentioned above (Reviewer #1), we point out that the current work also includes a Prosit model for (resonance-type) CID fragmentation which is entirely new.

To address the specific question of this reviewer, we added an additional Supplemental Figure S7 that details the prediction performance of the 2019 and 2020 model on the holdout dataset. The reviewers’ intuition was correct in that the 2020 model outperforms

the 2019 model in terms of prediction accuracy for singly charged peptides (Supplemental Figure S7a) and for peptides regardless of their C-terminal amino acid (Supplemental Figure S7b). The spectral angle distributions are much tighter for the 2020 model and lack the 'long tail of poor predictions' that the reviewer noticed for the 2019 model, especially for the non-tryptic peptides.

* On a related note, a general question about the magnitude of improvement in the ProSIT 2020 model is whether the comparison metrics are sensitive to different measures of spectral similarity, e.g., spectral angle encodes the entire spectrum and all peaks as a multidimensional vector and so would be more sensitive to minor differences, but most proteomics search engines don't score spectral similarity with spectral angle. What is the implication here and what would be the result if spectra were compared using other metrics, e.g., skyline dotp or sequest-style xcorr?

We (and others) use the spectral angle (SA) specifically to be able to distinguish small differences between (predicted and/or experimental) spectra. This is because the ability to detect small differences in fragment ion intensities, enables a better separation of correct and incorrect matches.

While the xcorr metric is frequently used in the field (implemented in the search engine Sequest), we were unable to find a reference implementation which would have allowed us to calculate xcorr values between predicted and experimental spectra. We refrained from implementing xcorr ourselves because we cannot ascertain that it would be the same as in Sequest.

We note that Skyline recently switched to using the normalized spectra contrast angle (see <https://skyline.ms/announcements/home/support/thread.view?rowId=25584>). Still, we calculated the dot product for a subset of our data, as for example MS PepSearch still uses this metric (new Supplementary Figure S6b). In addition, we added an analysis comparing the normalized SA to Pearson's correlation and the (normalized) dot product (dotp). For example, at a prediction accuracy of 0.9 SA, the corresponding Pearson correlation and dot product values are ~ 0.99 and ~ 0.99 , respectively. More generally, a spectral angle range of 0.8-1.0 (20% of its range) corresponds to a range of dotp and Pearson correlation of ~ 0.95 -1.0 (5% of their ranges), highlighting the superior sensitivity of SA to capture slight variations in spectral similarity.

* Does the ProSIT 2020 model also improve on predictions of tryptic peptides with internal basic sites (histidine or miscleavages) or chymotryptic peptide?

We have added additional analysis detailing the improvement of the 2020 ProSIT model compared to the 2019 ProSIT model in the new Supplementary Figure S7. Supplementary Figure S7d shows that the 2020 ProSIT model only exhibits a small increase in prediction accuracy for tryptic peptides (tested on 367,520 spectra of the hold out dataset) with one or

more internal basic residues. The median SAs are 0.889 (2020 Prosit model) vs 0.878 (2019 Prosit model). The fact that the 2019 Prosit model was able to predict such peptides with such accuracy was somewhat expected because the training data of this model contained a large number of peptides with up to two missed cleavages. This is because we had to systematically synthesize very many peptides for proteins that are never or rarely experimentally identified and these included many peptides with one or two missed cleavages. We called this set “Missing Gene” in the initial ProteomeTools publication (REF). Similarly, the holdout data contains 97,848 spectra that one could classify as chymotryptic peptides (Supplementary Figure S7c). Comparing the median SAs of these peptides showed that the 2020 Prosit model gained in prediction performance: 0.908 (2020 Prosit model) vs. 0.842 (2019 Prosit model). This result is in line with the data shown in Supplemental Figure S7b, where the prediction performance is detailed for peptides with different C-terminal amino acids.

2. The authors then reanalyzed the MHC-bound peptide data from Sarkizova et al. and found that rescoring the candidate PSMs using the 2020 Prosit-predicted spectral intensity improved identification (~50%) (main text line 181 and Supp Fig 11).

* Given the major novelty of the current study, is the improvement of the Prosit model through training of new non-tryptic peptides, a fairer comparison would be to compare the 2020 Prosit-based rescoring with similar rescoring using the 2019 Prosit model or other approaches that predict fragment ion intensities (e.g., MS2PIP, pDeep).

The comparison of the 2019 and 2020 versions of Prosit is shown in Figure 2b of the main manuscript and in the new Supplementary Figure S7a-d. From that it is clear that the Prosit 2020 model led to strong improvements, particularly for singly charged (1+) peptides. The latter are often found for HLA peptides. To illustrate this further, we rescored the allele C*08:02 using the 2019 and 2020 Prosit models (new Supplementary Figure 12b). Briefly, the results show that while the 2019 model led to a net increase of ~300 peptides, the 2020 model increases the number of identified peptides by ~850. In addition, the 2020 model also showed fewer losses compared to the 2019 model. This allele was picked as an example particularly for the reason that it binds HLA peptides that were predominantly detected as singly charged precursor ions. For other alleles, such as the tryptic-like A*11:01, we did not observe a major improvement of the new model.

3. The authors also re-analyzed data related to proteasome-spliced peptides and found a large proportion of false positives (87%; line 253). This promises to be a useful result and adds to the ongoing debate about the extent of these events.

* The Liepe et al. Science 2016 study attempted to validate 98 spliced peptides against synthetic peptides. Are any of those peptides captured in this reanalysis and do they show better Prosit prediction than other, un-validated peptides?

To clarify, we investigated the results reported by Liepe et al 2018, not 2016. Unfortunately, the 2016 study did not provide any information which would have allowed us to extract the experimental spectra of the synthetic peptides. In addition, the “validated” peptides of the 2016 study showed rather poor Pearson correlations (~0.5 on average; see original Figure from that study below). In our experience, such poor correlations are insufficient to validate identifications using synthetic peptides and we previously have employed cutoffs of SA >0.7 (Pearson R ~0.9) in our publications for the validation such spectra in Wang et al. 2019 (doi: 10.15252/msb.20188503).

Image from Supplementary information of Liepe et. al (2016); doi: [10.1126/science.aaf4384](https://doi.org/10.1126/science.aaf4384)

We had also contacted the authors of the 2016 study some time ago and were told that “...the analysis we [Liepe et al] did in 2016 is outdated. [...] In this [2018] paper we also show that we might have wrongly assigned some of the sequences in 2016 using the first SPI method.”. Unfortunately, only a single synthetic peptide represents one of the 1,230 proposed spliced peptides of the 2018 study which, unfortunately precludes any analysis of the type requested by the reviewer.

* In general I believe this section should be more precisely worded, as well as the Discussion line 320 (“concept of proteasomal [splicing] ...is not well founded”). The workflow here

arrived at the same conclusion of some previous analyses that many of the putative proteasome-spliced peptides have low search scores. This is consistent with a relative rarity of proteasome spliced peptides, but nevertheless the workflow reconfirmed ~3% of the HLA identified peptides to be spliced, some of which may be important for the search of cell-specific epitopes.

We were careful not to conclude that the 13% of peptides that remained after our re-analysis of the data are genuine spliced peptides. This is because we cannot claim that we have considered all possibilities for alternative explanations. As discussed further below, we found that the remaining proposed spliced peptides show a larger than expected (and statistically highly significant) difference between experimentally determined and predicted retention times. We would rather not strengthen the notion that proteasomal splicing may exist as the large majority of data currently suggests otherwise. The authors think that it is up to the proponents of the existence of such splicing mechanisms to unambiguously show that such events do indeed happen in cells.

* Are there scoring differences between the mechanistically more intuitive cis-splicing vs. trans-splicing?

Unfortunately, we could not investigate this in more detail. The 2018 study only mentioned trans-splicing once and the annotations available to us for the proposed spliced peptides do indicate that all of them are the (proposed) results of cis-splicing. Instead, we have added a bean plot of the Mascot ion score (new Supplementary Figure 13e) of the most recent (large scale) study performed by Specht et al. (Scientific Data 2020) which shows that most of the proposed spliced peptides show substantially lower scores and we did not observe major difference between cis, reverse-cis and trans splicing. This plot independently underscores our interpretation that the proposed spliced peptide events may be the result of artifacts of the data analysis pipeline. Again, the collective results do not provide clear evidence that proteasomal splicing exists.

* It would be useful to see if the false positive spliced peptides also have different experimental v.s predicted retention time (e.g., in Rolfs et. al. Sci Immunology 2019)

We have added an analysis of the (indexed) retention time (iRT; new Supplementary Figure 13d) which independently confirms the observation by Rolfs et al. (Sci Immunology 2019) in that the proposed spliced peptides exhibit larger RT differences (experimental vs predicted) than non-spliced peptides. Again, this strengthens our hypothesis that spectra matching to proposed spliced peptide may actually represent ordinary peptide sequences that are not in the database used for peptide identification.

* Line 248: "The Percolator score difference between the proposed spliced (Score = 2.9) and canonical peptide (Score = 2.5) is small." What are the Percolator q value and PEP for the spliced vs. non-spliced peptides, since Percolator already takes into account the scoring difference between the first and the second hit?

Unfortunately, the published data only provides the top ranking and FDR filtered target hits. Therefore, we cannot assess the difference in quality between the first and second hit. In our analysis, we used the combination of Prosit and Percolator to enable the direct comparison of results obtained from multiple search engines. Briefly, a single Percolator model was trained (on the rescored MaxQuant results) and subsequently used to align the results obtained from the other search engines. While the scores are well aligned, indicated by the PSMs where the same peptide sequence was identified by two search engines (Supplementary Figure S12a), we can only assess the q-value and PEP of the MaxQuant results in a fair fashion. Specifically, the q-values and PEPs of the non-spliced peptides was 0.0000116 and 10^{-16} , respectively. When extrapolating the q-value and PEPs from the MaxQuant results to Mascot, the proposed spliced peptide has the same q-values and PEPs (because no additional decoys were identified in this score range by MaxQuant). However, we note that the actual q-values and PEPs of the proposed spliced peptides by Mascot might be lower, given the significantly larger search space.

4. The authors reanalyzed the melanoma MHC1/MHC2 peptides in Bassani-Sternberg et al. Nat Comm 2016 and found additional peptides including neo-epitopes.

* There should be a more comprehensive audit on the nature of the gained identification via Prosit 2020 rescoring. For the spectra responsible for the gained identification, what sequences were they identified to in the original pipeline?

For clarification, we briefly summarize the concept of rescoring: As data input, we initially perform a MaxQuant/Andromeda search without FDR filtering on PSM or protein level. Hence, all identifications by MaxQuant/Andromeda – regardless of their score – are rescored by predicting the spectra of the candidate PSMs and calculating a variety of intensity based scores for the PSMs. Afterwards, the semi-supervised machine learning algorithm Percolator combines all scores into a single score which is used for FDR estimation. Hence, only spectra can be identified which were identified by MaxQuant/Andromeda in the first place. Therefore, rescoring never adds additional identifications but ‘rescues’ PSMs which the other search engine did not retain. As a result, and unfortunately, we do not have access of the unfiltered results of the original study. Therefore, we cannot perform the spectrum by spectrum comparison as requested.

E.g., were the best PSM identified to the same sequence but not passing FDR cutoff without rescoring? How close were the rescored vs. original PSM in terms of scoring? Regardless of whether the PSM newly passed Percolator PEP/FDR cutoff, I believe there should be an

attempt to quantify or at least visualize the gained explanatory power on a per-spectrum level, similar to what was done for spliced peptides in Supplementary File 2.

We can investigate the population of spectra which survive the FDR threshold after rescoring on the dataset at hand: As a consequence, we have added a more detailed analysis on the gained explanatory power provided by the rescoring pipeline. Briefly, we made two observations.

First, spectra uniquely identified by the rescoring pipeline had a lower total ion current (new Supplementary Figures S15d). This is in line with earlier observations that spectral similarity measures such as spectral angle are more stable across precursor intensities than scores based on counting fragment ion only. This is due to the fact that counting fragment ions (or probabilistic scores) strongly depend on the quality of a spectrum, because every single additional fragment ion observed has a positive effect on the overall score (matching 5 of 10 of the theoretical fragment ions is less confident than 6 of 10). This favors spectra with very high signal to noise and peptides which fragment into many intense fragment ions. If either is not the case, the resulting identification score is low. In contrast, this is not the case for spectral similarity metrics where the overall score is less dependent on the number of fragment ions.

This is supported by the second observation, showing that rescoring 'rescues' spectra that produce low identification scores by classic search engines (new Supplementary Figures S15a and c). This is expected because, otherwise, high-scoring PSMs would have survived the FDR cutoff in the first place. However, the 'rescued' identifications are located in a region of the score distribution, where one typically observes a significant enrichment of targets. This suggests that many potential true positives are missed due to suboptimal separation of correct and incorrect matches. The gained identifications show a scoring distribution that is nearly the same as the identifications shared between rescored and non-rescored data when compared to the rescored distribution. This overall characteristic was also observed for the mono-allelic data set (Supplementary Figure S12 c-d).

* Line 281, it is not clear why the Maxquant reanalyses here were able to reidentify substantially more (36 vs. 8) melanoma immunopeptides over the original Bassani-Sternberg et al. study, since Maxquant was also used in that study.

The authors think that the differences in the two MaxQuant processings lie within the patient specific fasta database used for processing the Mel15 patient. According to authors of the 2016 Bassani-Sternberg publication, they used "...newly designed module of the MaxQuant software that enables the search for peptides based on genomic variations". "[The] new module in the MaxQuant software that performs mutations calling from [genomic] NGS data and generates a customized personalized reference database containing all protein isoforms where a detected SNV alters the amino acid sequence".

For the reanalysis of the Mel15 data a new patient-specific database was used that now also contains mutations identified from RNA sequencing data (see Methods), thus including also non-coding transcripts and increasing the overall size of the patient specific database. This explains the increase in identified neo antigens by MaxQuant in this study compared to the original publication.

5. Overall it would be useful to see a more in-depth analysis of where the improvement in the Prosit 2020 prediction comes from and what features might be important for accurate MS2 prediction, e.g., what ions are given more intensities in the 2020 models the case of an internal basic residue. It is not surprising per se for a deep learning model to perform better when being fed substantially more data, but there is a great opportunity here to further dissect the black box of machine learning predictions.

To address these questions, the authors have added a new Supplementary Figure S7, which details differences of the two Prosit models for different peptide categories. As the reviewer correctly points out, the improvements can be attributed to the vastly increased training data for non-tryptic peptides and peptide precursors with a charge state of 1+. For an explanation for the slight improvement of prediction accuracy for peptides with internal basic residues, please refer to the comments above.

The authors agree that dissecting the black box of neural networks would be of very high interest. However, this poses a major challenge and has given rise to a whole new field of research (Explainable Artificial Intelligence or XAI). Recently, dimension reduction techniques such as UMAP or tSNE can be used to project the high dimensional latent space of neural networks into 2-D images. To interpret the resulting representations, such images can then be overlaid with features that are deemed important. While such approaches can work for simple problems, like iRT prediction where the resulting tSNE image of a model correlates well with the hydrophobicity of peptides; they may not provide a clear picture for more complex systems such as fragmentation models. Our current manuscript focusses on demonstrating the high prediction accuracy of the Prosit models as well as their application. At this time, we think that venturing into opening the black box of neural networks is beyond the scope of the current manuscript.

Reviewer #3 (Expertise: Neopeptidomics):

Wilhelm et al. have submitted a manuscript on improved prediction of MS/MS spectra of HLA ligands and non-tryptic peptides using previously established deep learning framework Prosit. The authors performed multi-modal LC-MS/MS measurements of over 300,000 peptides representing HLA class I & II ligands and products of the proteases AspN and LysN,

and used the resulting nine million MS/MS spectra to re-train the Prosit platform. They report significant improvements in prediction of MS/MS spectral features by the new version of Prosit, especially for singly charged, non-tryptic peptides, but also for higher charged peptides and other commonly used types of fragmentation, such as CID. As a result, Prosit is now significantly improved and well suited for immunopeptidomic applications across several MS platforms. The Authors demonstrate improvement in the number of HLA peptide identifications by over 25% after re-processing the MS/MS spectra obtained in the benchmark dataset of Sarkizova et al. They further apply the re-trained Prosit platform to demonstrate that about 90% of the “spliced” peptides previously reported by Liepe et al. are either wrong, or can be explained by canonical proteasomal processing. Finally, they apply Prosit to another landmark study of Bassani-Sternberg et al. that analyzed HLA-ligandomes in 25 melanoma patients; importantly, the Authors detect 7-fold more HLA class I peptides in this dataset and successfully follow up several of the novel HLA epitopes.

Just to clarify, while the Prosit framework and architecture is the same, the 2020 HCD and 2020 CID Prosit models are new.

In summary, this is a carefully executed large-scale MS study that significantly improves an established platform for prediction of MS/MS spectral features (Prosit). The re-trained platform has a potential to revolutionize immunopeptidomics, accelerate development of anti-cancer vaccines and personalized medicine in general. Moreover, the study addresses and challenges the controversial concept of proteasomal peptide splicing. This is the only part of the manuscript where I missed a more thorough analysis of the available datasets (more MS files), as well as more detailed treatment of the “surviving” spliced HLA peptides; e.g. it would be good to know what kind of scores/intensities they have because even this low number may point to the existence of HLA peptide splicing at very low levels or in a small population of cells. Although this part will surely incite a discussion, I see this is as a minor point and I fully recommend publication of the manuscript in Nature Communications.

The authors are happy to read that this reviewer sees the substantial potential of our work for the field of immunoproteomics. Regarding the specific point of spliced peptides, we now provide a more in-depth analysis as requested (please also see our comments above). We first investigated the retention time (RT) characteristics of the “surviving” proposed spliced peptides and found that - while to a lesser degree than the rejected spliced peptides - they still show a substantially higher RT difference between the prediction and the experimental values (new Supplementary Figure 14d). This supports our position that a large fraction - if not all - of the “surviving” peptides are likely also incorrect identifications as one would otherwise expect their RT deviation to be similar to the non-spliced peptides. For the following reason, we also think that neither the score (average Mascot ion score of 36) nor the intensity of the “surviving” peptides provides a reasonable indication of their

correctness. We have observed that the sequence of many of the proposed spliced peptides show minor differences to canonical sequences so their overall scores and intensities will be high, although incorrect because most of the fragment ions supporting either the spliced or an alternative sequence are the same (see our example in Figure 4c).

We added an analysis of the score distributions of non-spliced, cis, reverse cis and trans spliced peptides from a recent study (Specht et al. Scientific Data 2020, <https://doi.org/10.1038/s41597-020-0487-6>) and found that spliced peptides have a substantially lower median score compared to non-spliced peptides (new Supplementary Figure 14e). This also strengthens our view that most (if not all) proposed spliced peptides are more likely artifacts of the data analysis pipeline than genuine splicing events.

Reviewer #4 (Expertise: deep learning for mass spec analysis):

The authors synthesized and analyzed the LC-MS data of more than 300,000 peptides of HLA class I, HLA class II, and other non-tryptic peptides. The data was then used to train a deep learning model, Prosit, to predict MS/MS spectra of HLA peptides. The spectra prediction model improved the identification of HLA peptides by database search engines such as MaxQuant, MSFragger, and subsequently might lead to the identification of more neo-epitopes for cancer immunotherapy.

Just to clarify, the new Prosit 2020 model not only predicts spectra for HLA peptides, but any peptide sequence (7-30 amino acids) as this new model was trained on the combined ProteomeTools dataset (including tryptic and non-tryptic peptides). We also report a new (but separate) Prosit model that makes CID data accessible to spectral predictions.

Detailed comments:

The title is not clear. The title should briefly state the research problem/results of the paper. When reading the current title, I don't understand, for example, what "one mass spectrum at a time" refers to? In addition, both "deep learning" and "immunopeptidomics" are very broad topics, so it is not clear what specifically this paper is trying to do. I think the main contribution of this paper is the synthetic HLA peptides, so it should be reflected in the title.

We chose a relatively broad title to reflect that Prosit can be broadly applied. In that sense, mentioning "immunopeptidomics" in the title already highlights one particularly interesting area of application. 'One spectrum at a time' reflected the fact that Prosit re-scores every single candidate peptide identification and which is a key factor underlying the observed improvements. Together, we felt that the wording made the title interesting for both the proteomics informatics as well as the immunopeptidomics communities. Still, we have adjusted the title to "Deep learning boost sensitivity of mass spectrometry-based immunopeptidomics".

We respectfully disagree with the assessment that the synthetic HLA peptides are the main contribution of the paper. While this resource of reagents is certainly unique, its value is limited. This assessment stems from the fact that very few laboratories have requested a copy of any of the ProteomeTools synthetic peptide libraries over the past several years even though we have offered sharing these resources many times. In contrast, the LC-MS/MS data created for these peptides are extensively used by the community. In fact, the ProteomeTools data is among the most downloaded data from PRIDE (<https://github.com/bigbio/proteomics-metadata-standard/issues/66>), so it can be safely assumed that other scientists in the field make active use of the data. We anticipate that

this will continue to be the case for the data that will be released as part of this manuscript. The spectral libraries generated from the data are, unfortunately, also only of limited use. As we argue in the manuscript, the number of possible HLA peptides is far too large to be comprehensively covered by synthetic standards. Therefore, any analysis done with the spectral library itself will be incomplete and it may be argued that HLA peptides not discovered so far might be of highest interest. The authors further anticipate that, beyond the peptides and the data shared with the community, Prosit substantially improves the analysis of any non-tryptic peptide which is scientifically relevant and opens its application to additional areas of research. As mentioned above, the implementation of Prosit in ProteomicsDB (in 2019) has led scientists to request more than two billion predictions (equivalent to ~44,000 hours of MS time), demonstrating substantial impact on our field. We will roll out the 2020 Prosit HCD and CID models to ProteomicsDB as part of this publication which will increase the value of this service further.

Lines 55-56: “>300,000 peptides by multi-modal LC-MS/MS within the ProteomeTools project representing HLA class I & II ligands and products of the proteases AspN and LysN”
It’s better to clearly state the number of peptides for each class as in lines 94-95.

Unfortunately, the abstract is limited to 150 words which restricts the level of detail that can be included. The numbers are explicitly stated in both the results (including Figure 1a) and methods sections which provides the information in a transparent fashion.

Line 59: “the identification of HLA peptides can be greatly improved”
Please state specifically how much % the improvement was and for which database search engines.

We have rephrased the sentence on the improvement to read: “Applying Prosit demonstrates that the identification of HLA peptides can be improved up to 7-fold...”.
However, due to the word limit we could not add details about individual search engines.

Line 59: “proteasomal HLA peptide splicing may not occur to any noticeable extent”
This is not clear, please elaborate

We have also rephrased the sentence on HLA splicing to read: “...that 87% of proposed proteasomal HLA peptide splicing are incorrect”.

Line 102, Figure 1c: the column and the row corresponding to amino acid Cysteine (C) seem to have very low number of peptides, maybe the authors could provide more explanation why this is the case.

The short answer is that Cys residues are strongly underrepresented in the data sources that formed the basis for the selection of peptides to synthesize in the ProteomeTools project. In more detail, we assembled peptides from ProteomicsDB (proteotypic dataset), in-silico generated peptides for genes lacking mass spectrometric evidence (missing gene dataset), the HLA peptide databases IEDB, SystemMHC as well as prominent publications (HLA peptides and other non-tryptic proteases). As a result, 0.2 % of peptides contain a C-terminal cysteine and 0.9 % of peptides contain an N-terminal cysteine. Peptide synthesis or detection by LC-MS/MS did not introduce a noticeable bias (Supplemental Figure S1c shows >80% recovery rate for synthesized peptides containing a C-terminal cysteine). One factor may be the observation that many immunopeptidomics sample workflows skip the alkylating step. Under non-reducing conditions, unmodified cysteine residues are prone to form intramolecular disulfide bridges and may, therefore, escape detection. But even if alkylation was included (e.g. Bassani-Sternberg et al. 2016) only 82 of the identified HLA peptides contained a C-terminal or N-terminal cysteine.

Figure 1e: Could we have the distribution of Andromeda score for tryptic peptides for comparison? And the distribution of Andromeda score for endogenous HLA-I and HLA-II peptides (where data is available, for example, reference 11).

We have added both the distributions of Andromeda scores for tryptic peptides as well as the distribution of scores from reference 11 to the new Supplementary Figure S2b and S2c. The distribution of the scores of the HLA, HLA Class II, ApsN and LysN peptides are very similar to the tryptic peptides and using the different fragmentation methods (new Supplementary Figure S2b). This is not unexpected given the wide sequence variety of both datasets.

Concerning the comparison with data from endogenous peptides from reference 11 (new Supplementary Figure S2c), two things are apparent: First, the Andromeda score distribution of HLA Class I is tighter and has a higher median than the HLA Class II peptides. One of the reasons for the high median scores for HLA Class I peptides in the endogenous sample is the fact that MaxQuant's FDR algorithm needs to apply a high score cutoff to maintain 1% FDR. This leads MaxQuant to retain only high scoring identifications with a very similar median score to the synthetic peptide data for HLA Class I peptides. For HLA Class II peptides, the difference of the endogenous median score to the synthetic peptide is large. This could be due to the overall lower abundance of HLA Class II peptides and overall lower data quality of HLA Class II raw files in the reference 11 dataset.

Lines 138-141: "Combining the 9 million HCD spectra of non-tryptic peptides collected in this study with the 21 million spectra previously published tryptic peptide enabled the training (70% of all spectra), testing (20%) and validation (10%, referred to as holdout set) of a single high-performant ProSight model for both types of peptides (Figure 2a, Supplementary Figure S6a, Methods)"

The authors emphasized throughout the manuscript about “a single Prosit model” for both tryptic and non-tryptic peptides. While using a single model maybe more convenient, one could argue that specific models for each type of peptides would provide more accurate predictions. Could we have a comparison between a single model versus specific models? Given the amount of data available in this paper, I believe it is sufficient to train specific models. As the authors have mentioned, these two types of peptides are very different in fragmentation characteristics, so specific models are desirable. My other concern with the single model is how to deal with the class imbalance, e.g., the amount of tryptic spectra is more than double that of non-tryptic spectra, so the single model is definitely bias towards tryptic spectra after training.

Last but not least, HLA peptides are very diverse and depend on a large number of different HLA alleles. In the current manuscript, there is no analysis/discussion about HLA alleles of the training data. Furthermore, it would be interesting to explore if one could train a model specific to each HLA allele, as we are moving towards the direction of personalized immunotherapy.

There are several points to respond to here. For clarification, this manuscripts reports on two new Prosit models. One for beam-type HCD (fragmentation in a multipole) and one for resonance-type CID (fragmentation in an ion trap).

As for the single HCD model the reviewer refers to, the advantages of a single model go beyond convenience. That said, the element of convenience should not be discounted because a single model that serves most purposes with good performance will be of value for many users because it relieves users from the burden of choosing the best model for their purposes. Still, the authors do acknowledge that specific models for specific groups of peptides can be more powerful than a combined single model. As we noted in response further below, comparisons performed on the synthetic peptides from the ProteomeTools project suggest that the prediction accuracy is close to the upper bound of what may be possible with our prediction framework which indicates that the current single model is very performant.

Training more specific models may come at the cost of overfitting, may suffer from too little training data or may have a narrow utility, particularly when boundaries between peptide classes become somewhat vague. For example, is an HLA peptide ending with K tryptic or not? Should different models be trained for singly, doubly and triply charged peptides because they show very different fragmentation characteristics? Similar arguments apply to peptides which either do or do not contain a Proline because fragmentation N-terminal to Pro often dominates the tandem mass spectrum. We argue that training a model for each of such cases may leave users at a loss as to which model should be used for what purpose. As mentioned just above, not all HLA peptides are very different from tryptic peptides. As shown for allele A*11:01 (Supplementary File 1, page 81-88), the binding motif is very close to tryptic peptides and peptides bound by this epitope cannot be readily distinguished from them. In light of the above, we refrained from training separate models for tryptic and non-

tryptic peptides and the performance of the single model suggests that it is fit for most purposes.

Regarding imbalance/bias, the fact that the new (combined) Prosit model improves predictions for both tryptic and non-tryptic peptides shows that imbalance/bias does not appear to be an issue. We note that the spectral angle obtained by the new Prosit model is slightly higher for HLA class I peptides (SA of 0.915) than for tryptic peptides (SA of 0.901). If imbalance had been an issue, the improvement particularly for the non-tryptic peptides would not have been realized. With this in mind, we think that a combined model has several advantages, and, as mentioned earlier, makes it attractive for users, which is not just a matter of convenience. Using different prediction models within one analysis pipeline will require additional alignment steps to make sure that e. g. the spectral angles from different prediction models are comparable to each other and do not bias downstream analysis and interpretation.

The authors agree that a more detailed analysis of the HLA alleles would be interesting. Unfortunately, the list of HLA class I (and II) peptides compiled from public repositories is not complete or even consistent concerning information which HLA allele the data represents. To show that the new Prosit model performs well across the different alleles, we used data from a large monoallelic cell line study (Sarkizova & Klaeger et al. 2020) to annotate and classify HLA Class I peptides contained in the holdout-dataset of the ProteomeTools training data. We then assessed the spectral angle distributions for the peptides assigned to the different alleles and found no bias over all alleles with a median spectral angle of 0.92 across all 92 alleles (new Supplementary Figure 12a). Hence, it appears that training models for individual alleles is not required. This may even be detrimental when analyzing samples that contain more than one allele especially when alleles have overlapping binding motifs. Training performant models for each allele may also be difficult, at least for our architecture, because of limited training data.

Prosit was intentionally trained without any additional information regarding which biological context a peptide originates from. One might argue that (almost) any arbitrary peptide sequence can be classified as either tryptic, HLA, spliced or mutated, because this label solely depends on the reference database. We know from earlier work that Prosit works well for phylogenetically distant organism because most organisms use the same repertoire of amino acids to build proteins. The tandem mass spectrometer is also unaware of a peptides' origin so it should be expected that a well-trained model would perform well for most peptides. The authors are confident that this manuscript shows that the 2020 Prosit model generalizes very well to peptide sequences never exposed to the model (Figure 2b - spectra from the holdout set) and thus facilitates the identification of neo-epitopes and which can be experimentally validated.

Regarding the training-validation-testing 70-20-10 partition, from my understanding, this partition is based on the spectra. However, different spectra may share the same peptide sequence, since we have near 30 million spectra but less than 1 million peptides. So it is possible that the training, validation, and testing sets may share common peptides. Deep learning models like Prosit may simply memorize those common peptide sequences from training and apply to testing, so the accuracy may not be accurate because we need to test on peptide sequences that the model has not seen before.

To clarify, the partitioning was done on the level of peptides, not spectra. Lines 409-410 in the initial submission state: "HCD [CID] Training data was split into three distinct sets with each peptide sequence only included in one of the three".

Lines 159-161, Figure 2e: Figure 2e has little meaning because it is certainly that training accuracy is always better than testing accuracy. One can compare different models on the same dataset, or different datasets on the same model (like Figure 2b), but training versus testing of the same dataset does not show anything interesting.

We agree with the reviewer that one should expect that the accuracy on the training set is better than that of the testing set, but this is not the reason why we show this figure panel. The authors think that it is imperative to demonstrate that the new model is able to accurately predict the fragment intensities of peptides which it has never seen before (holdout set or test set, not used during training at all) and that it did not "simply memorize [...] peptide sequences from training". Furthermore, determining the accuracy of the holdout set is important to be able to assess any new or unexpected peptide sequence (e.g. the much debated spliced peptides). This figure (and Supplementary Figure S6a) also highlights that training was stopped when no further progress was achieved on the validation dataset. This is important because although the loss on the training data is still decreasing, early stopping leads to a model which does not show major signs of overfitting.

Lines 172-177, Figure 3a: It would be interesting to show how much improvement in spectra prediction would lead to how much improvement in identification, especially for HLA peptides. For instance, is SA=0.90 is good enough for identification, or is it worth to try to reach SA=0.99 with more data or larger models?

This is a very interesting point and we did not have enough space in the manuscript to elaborate on this. We have estimated the upper bound of prediction accuracy. For that purpose, we compared different experimental HCD spectra from the same precursor (peptide/charge combination) from the ProteomeTools project and found that the average spectral angle is ~ 0.92 . This suggests that a prediction accuracy of SA 0.99 for all peptides/spectra cannot be (meaningfully) achieved. This upper bound is higher than previous estimates (Li et al. Analytical Chemistry 2012, doi: 10.1021/ac102272r), since the

quality of spectra from synthetic peptides (ProteomeTools) is typically much higher than spectra collected from biological samples. One reason for that is that the signal intensity in the tandem mass spectrum strongly depends on the abundance of the precursor ions. We think that the answer to “Is 0.9 good enough for identification?” depends on the context of the analysis. For example, comparisons of Prosit and MS2PIP performed earlier for tryptic full proteome analysis showed that a spectral angle of ~ 0.7 (Pearson correlation of ~ 0.9) appears to be sufficient. However, when the peptide sequence search space increases, i.e. for HLA peptides, the number of similar peptides competing for a spectrum increases and thus more accurate predictions are required to rank the different peptide hypotheses correctly and to allow confident separation of correct and incorrect identifications. Prosit is already very close to the estimated upper bound and thus we argue that Prosit will prove to be very useful for many (future) applications. Those include the analysis of DIA data and, perhaps even more so, targeted analysis by PRM. We note here that we will make the 2020 Prosit model also available in Skyline (which currently uses the 2019 model) where high accuracy is mandatory for targeted assays.

Lines 191-194: I don't quite understand. The authors compared MaxQuant standalone versus MaxQuant+Prosit. So how was the loss of 15,000 peptides related to SM HLA v2? It is better to show the score histograms of MaxQuant results versus MaxQuant+Prosit and highlight the common, extra, and loss peptides to see how confident those identifications are.

The outputs of different search engines can vary quite substantially. As Prosit rescoring relies on prior search engine results, the loss of the 15k peptides is due to MaxQuant not providing these identifications. Thus, the rescoring cannot rescue those (new Supplementary Figure 12c). The analysis showing how many peptides are common, extra and lost between MaxQuant and MaxQuant+Prosit is shown in new Supplementary Figure S11b. As expected, the additional peptides often have low Andromeda scores, which do not survive the FDR criterion used by MaxQuant. This is because the Andromeda score displays a low separation power especially in the region of the score distribution where targets and decoys overlap. The fact that that the number of targets is much higher than the decoys in this region shows that there are many true positive identifications, which cannot be confidently called by MaxQuant. As can be seen in the plots, the spectral angle adds separation power which leads to the ability to call more of the correct PSMs. Conversely, the lost peptides have very low spectral angles and are, therefore, deemed non-significant by percolator.

In the main manuscript, we first wanted to show that our previously developed workflow using MaxQuant + Prosit already improves on the status quo (i. e. the published Spectrum Mill analysis). We could ascertain this improvement by the fact that the gained peptides have very similar peptide motifs as the previously identified ones. We then moved on to

combining Spectrum Mill + Prosit. This was motivated by the fact that Spectrum Mill appears to be more powerful for the identification of HLA peptides as it identified many more peptides than MaxQuant. This led to the hypothesis that combining Spectrum Mill and Prosit could lead to further improvements. This turned out to be the case.

I think it is better to perform this performance comparison on the synthetic HLA peptides of this manuscript.

To clarify, we trained and validated Prosit using synthetic peptides. Applying the rescoring workflow to the same peptides is a circular argument and would not help to convince our readership that Prosit is applicable to other datasets.

Because otherwise, Prosit was already published in a previous paper and so was the dataset of 95 monoallelic cell lines, so there is little novelty here. I believe the main contribution of this manuscript is the dataset of synthetic HLA peptides and the analysis should focus on this dataset, rather than on Prosit and other things (which could be placed in the supplementary).

Please also see our comments above regarding what the authors think this manuscript contributes in terms of novelty. To clarify, the previously published Prosit model is not the same as the one presented in this manuscript. In addition, extracting a large number of additional HLA peptides from previously published experimental data constitutes substantial novelty. This is particularly true for the re-analysis of data collected from tumor patients and which led to the identification and validation of a new neo-epitope. None of the analysis shown (i.e. on the spliced peptides) would have been possible at this accuracy without the new Prosit model. Taken together, we are confident that the combination of the synthetic peptides, the MS data, training accurate models and using these to re-score a wide variety of datasets are substantial novel contributions of this manuscript.

Lines 226-230: I'm not sure whether this comparison is fair. The Prosit model was trained on non-spliced peptides, so one could expect its performance on spliced peptides was not as good as on non-spliced peptides. Particularly, Prosit is a deep learning sequential model, so it may memorize non-spliced peptide sequences but it has not seen any spliced peptide sequences during training. I would suggest the authors to re-train Prosit model on a fraction of spliced PSMs and test it on the remaining spliced PSMs.

Please see our comments above. We have shown that Prosit does not show any major signs of overfitting (Supplementary Figure S7a and S7d). There is also nothing special about (alleged) spliced peptides compared to other non-tryptic or HLA peptides as the proteases in the proteasome have trypsin-like and chymotrypsin-like cleavage specificities and these are covered in the training data. Also as noted further above, Prosit works well for species that

are very distant from humans (e. g. Arabidopsis, Mergner et al Nature 2020) although sequence conservation is very low between these species. Hence, one should not expect a model specifically trained on spliced peptides to perform any different, let alone better. More importantly, we think that re-training Prosit on a fraction of spliced PSMs is not meaningful or even possible. 1) It is unclear whether spliced peptides actually exist or not. Any collection of training data containing such spectra would currently have to be viewed with skepticism. 2) Even if these existed, there is not enough (ground truth) training data for spliced peptides on which a model could be trained. 3) Prosit is not aware of the presumed differences between spliced and non-spliced peptides and thus it would be very unlikely that it would be able to distinguish those.

Prosit only allows the rescoring of results of such search engines but is not a search engine itself. As mentioned at the beginning of this section, the results of search engines differ both in terms of calling correct and incorrect peptide sequences. Therefore, we used two search engines, MaxQuant and MSFragger, to maximize the number of peptides that can be considered. As the results show, there are many cases where canonical peptide sequences are the better matches to PSMs than alleged spliced peptides, again underscoring that training models based on genuine spliced peptides will not be possible.

In general, I don't think a universal Prosit model should be applied anywhere to any dataset, as I have mentioned above for tryptic and non-tryptic peptides. For example, the performance of non-spliced peptides in Figure 4a (SA=0.87) is worse than the results in Figure 2b (SA=0.92), despite that they are both HLA non-spliced peptides. So the model performance depends on the training data, and re-training is desirable when dealing with a new type of data such as spliced peptides.

Without repeating all the arguments, the authors respectfully disagree with the statement that "...no universal Prosit model should be applied anywhere to any dataset...". We have already demonstrated that a general Prosit model is able to increase the confidence and identification rate in classical proteomics datasets, as shown in the publication of 2019 and this study. The reason why the prediction performance appears worse for these non-spliced peptides is twofold. First, the list of non-spliced peptides contains false positive IDs, which decreases the average spectral angle. Second, and more importantly, the comparison between the two spectral angle distributions (synthetic vs experimental) cannot be easily done. The quality of experimentally acquired spectra strongly depends on the precursor abundance of the peptide at the time it was picked for fragmentation. As a result, fragmentation spectra of endogenous HLA peptides often have low signal to noise. Slight variations in the absolute abundance of fragment ions can have strong effects on the relative intensities. For training, testing and validation we used the highest scoring spectra (high signal to noise) acquired from synthetic peptides (high abundance). In addition, we note that the spectral angle is a very sensitive measure of spectrum similarity. For example, spectral angles of 0.87 and 0.92 correspond to a Pearson correlation of ~ 0.98 and ~ 0.99 ,

respectively. This difference can be explained by the difference in spectral quality (signal intensity) of synthetic and endogenous peptides and is not a result of a universal Prosit model being a poor predictor.

Apparently the binding motifs of spliced peptides are different from those of non-spliced peptides, and so are their amino acid compositions and hence fragmentation characteristics. Subsequently, I think it is not appropriate to use a Prosit model that was trained on non-spliced peptides to re-score database search results of spliced peptides from MaxQuant and/or MSFragger. So I don't think the authors' analysis and argument against proteasomal splicing are convincing.

Please also see our comments above. In addition, if the proposed spliced peptides were real, one would expect that their binding motifs were also the same as (or at least similar to) the binding motif of non-spliced peptides, as they would otherwise not be presented by the same (set of) receptors. In addition, as we have discussed earlier, Prosit is not aware that a peptide is spliced or non-spliced. In fact, the authors would be surprised to see if one could tell spliced from non-spliced peptides apart, if only the peptide sequence were provided. The label "spliced" and "non-spliced" is solely dependent on the reference database used. As a thought experiment, we may take another organism and would certainly not find a large majority of the synthesized HLA peptides. However, some of those "unmapped" peptides could be generated by potential splicing events in that organism. Using this new mapping, Prosit was in fact trained on a dataset containing spliced and non-spliced peptides. We argue that the classification "spliced" and "non-spliced" has no practical implication on Prosit's prediction accuracy.

Lines 273-275, Figure 5a: Please show the score distribution of newly identified PSMs superimposed on that of existing PSMs to see how confident the new identifications are.

The new Supplementary Figure panels S15a and S15c show the Andromeda and Spectral angle score distributions of the PSMs added, shared and lost by rescoring with Prosit. As expected, the PSMs added by Prosit are predominantly located in the lower Andromeda score region, hence not surviving the FDR cutoff. However, the PSMs added by Prosit have an almost identical Spectral angle distribution as PSMs shared between the new analysis, suggesting that the large majority of them are in fact true positives and could simply not be confidently identified by Andromeda's probabilistic scoring scheme.

Lines 279-284: I believe the authors of reference 11 (Bassani-Sternberg et al.) used MaxQuant in their proteogenomic analysis and identified only 8 neo-epitopes. Yet the current manuscript also used MaxQuant and identified 36 neo-epitopes. Please explain why there is such a big difference, did you use the same FDR threshold.

Please also see our answer to reviewer #2. Briefly, a new patient-specific data base was used that now also contains mutations identified from RNA sequencing data with subsequent improved mutation calling. This increased the overall size of the patient specific data base.

Reference 11 used MaxQuant 1.5.3.2, while this study used MaxQuant 1.6.1.0 for generation of the discussed results. Both studies used a PSM FDR of 1% and no FDR control on protein level for MaxQuant processing.

Please also show the identification scores, FDR, binding affinity, and immunogenicity of those neo-epitopes (similar to Table 1 of reference 11)

Please see Supplementary Table 5 tabs “Mel15 Results MaxQuant” and “Mel15 Results Rescoring” where we detailed the identification score, Q-values or PEP-values for all identified peptides and neo-epitopes. Analogous to Table 1 of reference 11 (Bassani-Sternberg et al., 2016), we have added the NetMHC4.0 and HLAthena binding affinity prediction for all identified neo-epitopes in both workflows for the alleles HLA-A*03:01, HLA-A*68:01, HLA-B*27:05 and HLA-B*35:03. Briefly, 23 of the 36 neo-peptides identified by MaxQuant in conjunction with the new database and 38 out of 78 neo-epitopes identified in the rescoring workflow were predicted to strongly bind to one or more of the HLA alleles. The results provided by predictions have been extensively discussed in the literature. It has been observed that predicted and experimentally determined binding affinities can differ greatly and that peptides predicted to be strong binders cannot necessarily be experimentally verified, and vice versa (doi: 10.1158/2326-6066.CIR-18-0584, doi: <https://doi.org/10.1038/s41598-020-77466-4>).

For the immunogenicity of all tested peptides, please see Supplementary Table 5, Tab “Immunogenicity assay” for the underlying acDC response assay data.

Lines 288-291: How is the immune response of the new neo-epitopes compared to those in the original study (reference 11)?

The immune responses of the new neo-epitopes presented in this study cannot be directly compared to previous neo-epitopes as the ELISpot assay is a qualitative screening assay and not a quantitative readout. Therefore, it just categorizes whether a peptide is immunogenic or not. Differences in response can be due to several different factors: T-cell frequency at the tested time point or within the tested aliquot, the T cell proliferation capacity in general, the clonal composition of the tested sample etc. that can differ within several assays, peptides, and time points. Therefore, responses between different experiments cannot be directly compared using the described stimulation method followed by ELISpot analyses.

The peptide “RTYSLSSALR” overlapped with the peptides reported recently by Tran et al. on the same patient Mel-15 dataset (<https://www.nature.com/articles/s42256-020-00260-4>),

where it was suggested to belong to HLA-II. Apparently the binding motif, 2nd and 9th amino acids (T and R, respectively), do not fit into any HLA-I alleles of patient Mel-15 (Figure 1b of reference 11). In contrast, Tran et al. reported multiple HLA-II peptides encompassing this mutation.

The peptide “RTYSLSSALR” shows in fact very good binding predictions for two different HLA-A alleles based on 3 different algorithms as provided in the table below. Thus, we strongly believe that this peptide should bind at least to one of the HLA-I alleles from Mel15 and is therefore a true HLA-I peptide.

Prediction of ligand RTYSLSSALR to all HLA-A and -B types of Patient Mel15

Algorithm	Affinity prediction HL*A-A03:01 (nM; %rank, BinderType)	Affinity prediction HL*A-A68:01 (nM; %rank, BinderType)	Affinity prediction HL*A-B27:05 (nM; %rank, BinderType)	Affinity prediction HL*A-B35:03 (nM; %rank, BinderType)
NetMHC 4.0	29.76; 0.15; SB	20.21; 0.3; SB	5001.44; 5.5	46131.8; 44
NetMHCpan 4.1 a	29.89; 0.103; SB	32.5; 0.430; SB	3981.8; 4.4	41258.3; 29.5
NetMHCpan 4.0	36.7; 0.1304; SB	38.2; 0.4445; SB	5305.2; 5.53	40273.9; 26.42

Also, the binding motif (2nd T and 9th R) does indeed fit into the binding patterns of HLA-A*03:01 and HLA-A*68:01, although it is not the most prominent motif identified for these HLA alleles. As in the Bassani-Sternberg & Bräunlein et al. publication only the motifs for 9mer peptides are shown, we also reanalyzed the GibbsClustering for 10mers in this study using the MHC Motif Decon Server and the identified motifs for the respective HLAs look very similar (see figure below) to those of 9mer peptides.

Motif clustering of 10mer peptides from Mel15 – used method: MHC Motif Decon Server, DTU

Nevertheless, in addition to the identified peptide “RTYSLSSALR”, also longer versions of this peptide harboring the same mutation might exist and thus would fit better to HLA-II molecules, as reported by Tran et al.

I found the number of 78 neo-epitopes of patient Mel-15 quite mind-boggling. Figure 5a shows the number of peptides of patient Mel-15 increased by 17,094 on top of existing 23,654, i.e. about 72% improvement. Yet the number of neo-epitopes increased by 10 folds (78 versus 8). So please list all the neo-epitopes, their identification scores and PSMs, etc.

For detailed identification scores, Q-values or PEP-values of the neo-epitopes, the authors would kindly refer the Supplementary Table 5, Tabs “Mel15 Results MaxQuant” and “Mel15 Results Rescoring”.

The sheer number of HLA peptides and neo-epitopes identified from this patient is indeed remarkable and which is why several publications have resulted from the analysis of this particular HLA repertoire. To clarify, we arrived at the number of 78 neo-epitopes in two steps. First, in contrast to the original publication, we used a patient-specific database built from whole exome and RNA sequencing data with subsequent mutation calling of that patient. Searching this sequence list using MaxQuant led to the identification of 36 neo-epitopes (see Figure 5b). Second, Prosit rescoring increased this number to 78. Hence, the effect of Prosit rescoring is a 2.2-fold improvement which is similar to the 1.7-fold increase obtained for the total number of peptides.

Reviewers' Comments:

Reviewer #1:

Remarks to the Author:

The author has seriously and constructively responded to all the questions I raised during the review. I appreciate these responses and no more questions. I suggest that the journal accept and publish this paper.

Reviewer #2:

Remarks to the Author:

The authors have done a comprehensive job in response to my previous comments. The new manuscript includes a number of new analyses and figures that addressed my major suggestions, including New Supplementary Figure S6b (dots vs. SA), Supplementary Figure S7c (prediction of peptides with internal basic residues), Supplementary Figure S13b (comparison of prediction improvements between Prosit 2019 and 2020 on HLA peptides), Supplementary Figure 15d (properties of rescued peptides)

- A minor comment I have is that the authors suggested they could not find a reference implementation of xcorr in response to my previous comment (Comment 1 part 2). This is somewhat puzzling as xcorr calculation has been well described in the literature since the 1990's including some modern variants such as by Noble, MacCoss, and others. Other spectral comparison methods such as used by MS-GF+ are also well described that in theory could be evaluated to see whether the new Prosit model improves spectrum similarity when other metrics than spectral angle are used. However this is not a major issue as the authors already include a new Supp. Figure S6b that compares dots to SA.

- Re: updated article title, "boost" should read "boosts" title. Also the title should also be updated in the supplementary file to reflect the change.

- Lastly I read with interest Reviewer 4's comments about whether Prosit analysis alone can exclude proteasome peptide splicing. In theory this could be done by synthesizing a subset of the high-confidence purportedly proteasome-spliced peptide sequences and showing that Prosit does predicts the synthetic peptide spectra well whereas the spectra for the purported identification in the original studies differ substantially from the predicted/synthetic spectra. I remain convinced that sections of the main article concerning peptide splicing could be more precisely worded. That a machine learning model poorly predicts some purported peptide-spliced sequences is to me not the same as demonstrating evidence that they are incorrect. Likewise the "validity and utility" of peptide-spliced sequence database should be up to the community to decide. While I agree with the authors that the extent of these events are most likely highly inflated, rarity does not necessarily imply a lack of utility.

Reviewer #3:

Remarks to the Author:

The authors have addressed all of my comments and I endorse publication of the manuscript in its current form.

Reviewer #4:

Remarks to the Author:

The authors have fully addressed all of my concerns.

REVIEWERS' COMMENTS

Reviewer #2 (Remarks to the Author):

The authors have done a comprehensive job in response to my previous comments. The new manuscript includes a number of new analyses and figures that addressed my major suggestions, including New Supplementary Figure S6b (dots vs. SA), Supplementary Figure S7c (prediction of peptides with internal basic residues), Supplementary Figure S13b (comparison of prediction improvements between Prosit 2019 and 2020 on HLA peptides), Supplementary Figure 15d (properties of rescued peptides)

The authors are happy to learn that the reviewer is satisfied with the efforts taken to revise the manuscript.

- A minor comment I have is that the authors suggested they could not find a reference implementation of xcorr in response to my previous comment (Comment 1 part 2). This is somewhat puzzling as xcorr calculation has been well described in the literature since the 1990's including some modern variants such as by Noble, MacCoss, and others. Other spectral comparison methods such as used by MS-GF+ are also well described that in theory could be evaluated to see whether the new Prosit model improves spectrum similarity when other metrics than spectral angle are used. However, this is not a major issue as the authors already include a new Supp. Figure S6b that compares dots to SA.

'In theory', this would be possible. That said, none of the tools and groups mentioned actually provide a library or software that one can use to feed two spectra and obtain an xcorr as an output. Therefore, the authors would have had to re-implement xcorr in a separate software tool. While this would also – in principle – be possible, the authors could not ascertain if our implementation would be the same as that in e. g. Sequest or other search engines. Therefore, we did not look into xcorr any further. We hope that the reviewer may find the comparisons provided in the manuscript acceptable.

- Re: updated article title, "boost" should read "boosts" title. Also the title should also be updated in the supplementary file to reflect the change.

We have changed the title and the supplementary files accordingly.

- Lastly I read with interest Reviewer 4's comments about whether Prosit analysis alone can exclude proteasome peptide splicing. In theory this could be done by synthesizing a subset of the high-confidence purportedly proteasome-spliced peptide sequences and showing that Prosit does predict the synthetic peptide spectra well whereas the spectra for the purported identification in the original studies differ substantially from the predicted/synthetic spectra.

The authors agree that adding spectra of synthetic peptides could have added to the discussion. At this stage, the journal editors advised us not to perform additional experiments. Instead, we have added text to the main manuscript that picks up this point. Briefly, researchers proposing to have identified spliced peptides should not only rely on the output of search engines, but also test the

candidates against predicted spectra as well as synthetic peptide standards. The latter has indeed been done in the literature but the interpretation of this data is often difficult as it is not very clear what level of similarity of synthetic or endogenous peptide spectra would be sufficient to call the two spectra the same. Adding a third (predicted) peptide spectrum will help but this will not overcome the basic issue. If predicted and synthetic peptide spectra are not extremely similar to the spectrum of the endogenous peptide, this information can be used to dismiss an e. g. spliced peptides but it will likely still be insufficient to call the spliced peptide with certainty. For that, orthogonal evidence supporting the identification of a spliced peptide would be required. Therefore, we argue in the manuscript (as others in the field would too) that one should first try to find alternative (more canonical) explanations for a spectrum before concluding that it indeed represents a splicing event.

I remain convinced that sections of the main article concerning peptide splicing could be more precisely worded. That a machine learning model poorly predicts some purported peptide-spliced sequences is to me not the same as demonstrating evidence that they are incorrect.

Please also see our comment above and we have revised the text accordingly. While the accuracy of Prosit in predicting spectra is very high, it is of course possible that it may make a mistake in individual cases and where the result of prediction would not be enough to dismiss a possible spliced peptide. As mentioned above, a spectrum of a synthetic peptide may reveal such a mistake. But to clarify, we did not aim to show that the spliced peptides are wrong, but we tested if there is a better explanation for a given spectrum in canonical peptide space. The revised text provides a more balanced discussion on this point and concludes that proteasomal splicing may occur, but that additional evidence is required to substantiate such an event.

Likewise the "validity and utility" of peptide-spliced sequence database should be up to the community to decide. While I agree with the authors that the extent of these events are most likely highly inflated, rarity does not necessarily imply a lack of utility.

We agree and have revised that part of the text.